# NoLan: Mitigating Object Hallucinations in Large Vision-Language Models via Dynamic Suppression of Language Priors

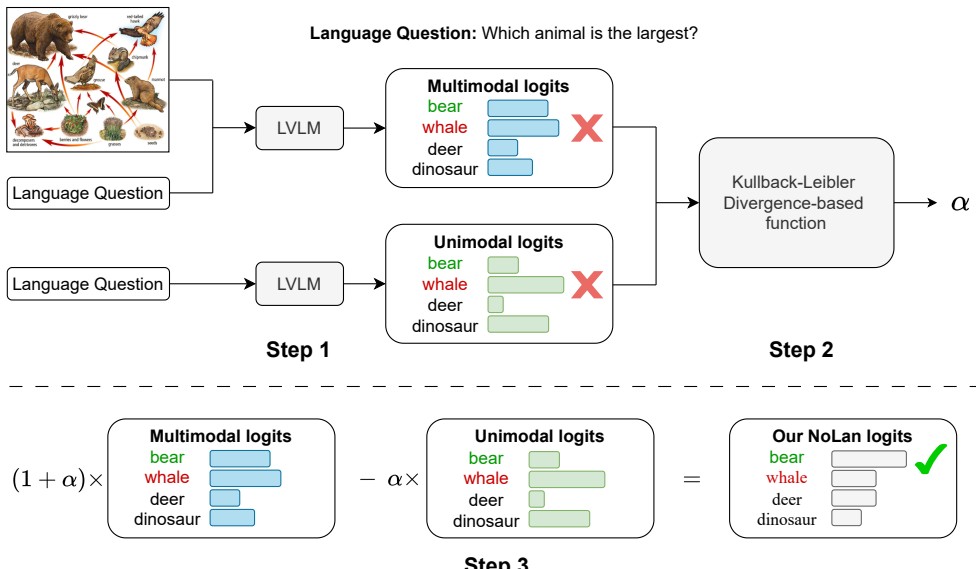

Figure 1: **No-Language-Hallucination Decoding (NoLan).** Given an LVLM, an image $v$, and a language question $x$, NoLan mitigates hallucinations in responses by comparing outputs generated from multimodal and unimodal (text-only) inputs. Step 2 can also be simplified by setting $\alpha$ to a fixed value of $1$. In this example, the hallucinated object "whale" is suppressed by reducing the influence of language priors during token generation, while the ground truth object "bear" is effectively enhanced.

## ABSTRACT

Object hallucination is a critical issue in Large Vision-Language Models (LVLMs), where outputs include objects that do not appear in the input image. A natural question arises from this phenomenon: Which component of the LVLM pipeline primarily contributes to object hallucinations? The vision encoder to perceive visual information, or the language decoder to generate text responses? In this work, we strive to answer this question through designing a systematic experiment to analyze the roles of the vision encoder and the language decoder in hallucination generation. Our observations reveal that object hallucinations are predominantly associated with the strong priors from the language decoder. Based on this finding, we propose a simple and training-free framework, No-Language-Hallucination Decoding, *NoLan*, which refines the output distribution by dynamically suppressing language priors, modulated based on the output distribution difference between multimodal and text-only inputs. Experimental results demonstrate that NoLan effectively reduces object hallucinations across various LVLMs on different tasks. For instance, NoLan achieves substantial improvements on POPE, enhancing the accuracy of LLaVA-1.5 7B and Qwen-VL 7B by up to 6.45 and 7.21, respectively. The code will be made publicly available.

# 1 INTRODUCTION

In recent years, Large Language Models (LLMs) (Touvron et al., 2023; Chiang et al., 2023b; Chen et al., 2023a; Zhao et al., 2023; Li et al., 2023e; Wei et al., 2023; Xiao et al., 2023) have revolutionized the field of machine learning with the ability of language understanding and content generation, offering unprecedented capabilities and potentials across a multitude of applications. The integration of LLMs with computer vision systems has given rise to Large Vision-Language Models (LVLMs) (Bubeck et al., 2023; Touvron et al., 2023; Zeng et al., 2022; Awadalla et al., 2023; Yang et al., 2023; Liu et al., 2023e; Zhu et al., 2023a; Ye et al., 2023b; Liu et al., 2020; Li et al., 2023c; Tran et al., 2020; Liang et al., 2023), facilitating various applications through their capacity to produce contextually accurate textual outputs from visual data. These models excel in identifying and converting intricate visual patterns into seamless linguistic expressions (Liu et al., 2023e; Zhu et al., 2023b; Ye et al., 2023a; Li et al., 2023a; Dai et al., 2023; Gong et al., 2023; Maaz et al., 2023; Zhang et al., 2023; Bai et al., 2023b). LVLMs with these advanced capabilities have demonstrated their value across multiple domains, such as content generation, image and video annotation, and interactive platforms that require comprehensive visual content interpretation. The development of LVLMs is characterized by continuous enhancements in model structures, training strategies, and data variety, resulting in improved performance and broader application adaptability. Nevertheless, a significant challenge persists: object hallucinations (Li et al., 2023d; Gunjal et al., 2023; Liu et al., 2023b; Lovenia et al., 2023), where the text generated by LVLMs does not accurately reflect the objects in the provided image. Object hallucinations can lead to misinformation and misinterpretation, posing significant risks for decision-making—particularly in high-stakes areas such as robotics (Mai et al., 2023; Liu et al., 2023c), autonomous systems (Chen et al., 2023b; Wu et al., 2023), and healthcare (Wang et al., 2023d; Hu et al., 2023).

In light of this, various strategies have been investigated to mitigate object hallucinations in LVLMs. Initial efforts focused on small-scale VLMs, employing techniques like fine-grained modality alignment (Biten et al., 2022) and data augmentation to reduce statistical biases related to object co-occurrence (Rohrbach et al., 2018; Kim et al., 2023). However, the distinct behaviors of LVLMs render these methods difficult to generalize and scale (Kaplan et al., 2020; Wei et al., 2022). Recent research has tackled this challenge by developing hallucination-specific datasets for fine-tuning (Liu et al., 2023a; Gunjal et al., 2023), training post-hoc revisors to produce outputs with fewer hallucinations (Zhou et al., 2023), and employing factually enhanced Reinforcement Learning from Human Feedback (RLHF) (Sun et al., 2023). Despite their effectiveness, these interventions demand significant human effort and computational resources, underscoring the urgent need for a simpler yet efficient solution.

LVLMs generally comprise two main components: a vision encoder that perceives visual information and a language decoder that generates text responses. This model composition motivates us to analyze the contributions of the vision and language components within LVLMs to the occurrence of object hallucinations. Through a series of analytical experiments, we find that object hallucinations primarily stem from the language decoder's priors rather than the vision encoder. Based on this insight, we focus on overcoming language priors and introduce *No-Language-Hallucination Decoding* (*NoLan*), a simple, effective, and training-free framework designed to mitigate hallucinations in LVLMs. As illustrated in Figure 1, NoLan works by contrasting the output distributions of multimodal inputs with those of text-only inputs, acting as a corrective mechanism to address the model's over-reliance on linguistic priors embedded in the LLM. The modulation of the output distribution increases when the similarity between the token distributions of multimodal and text-only inputs is higher, as measured by a Kullback-Leibler divergence-based function. Compared to previous methods (Liu et al., 2023a; Gunjal et al., 2023; Zhou et al., 2023; Sun et al., 2023), NoLan eliminates the need for additional training or external tools, such as other pre-trained models. Our experimental results validate the effectiveness of NoLan, demonstrating consistent improvements across various object hallucination benchmarks and LVLM families, including LLaVA-1.5 (Liu et al., 2023e;d), InstructBLIP (Dai et al., 2023), and Qwen-VL (Bai et al., 2023b). Specifically, on the POPE benchmark (Li et al., 2023d), NoLan achieves significant performance gains, with accuracy improvements of up to 8.38 and F1 score enhancements of up to 8.78, highlighting its robustness and scalability in addressing object hallucinations across diverse LVLM architectures.

Overall, our main contributions are as follows:

1. We conduct a series of analytical experiments to investigate the contributions of each component in LVLMs to object hallucinations, finding that hallucinations mainly stem from the language model's priors rather than the vision model.

2. Building on this insight, we introduce NoLan, a plug-and-play approach designed to mitigate object hallucinations by dynamically suppressing language priors. NoLan achieves this by leveraging the differences in output distributions between multimodal and text-only inputs, ensuring more consistent and contextually accurate content generation.

3. Extensive experiments demonstrate the effectiveness of NoLan in significantly reducing object hallucinations. Notably, our methods do not require additional training or external tools.

## 2 RELATED WORK

### 2.1 VISUAL-LANGUAGE MODELS

The evolution of Vision-Language Models (VLMs) has advanced significantly, shifting from language models that incorporate BERT-like language encoder (Devlin et al., 2018; Liu et al., 2019; Koroteev, 2021) for the fusion of visual and textual information (Li et al., 2019; Sun et al., 2019; Wang et al., 2022; Li et al., 2022a) to being driven by the integration of LLMs (Gilardi et al., 2023; Touvron et al., 2023; Tay et al., 2022; Raffel et al., 2020; Brown et al., 2020; Chowdhery et al., 2022; Taori et al., 2023; Chiang et al., 2023a; Bai et al., 2023a). By integrating a general vision encoder with a large language model, LVLMs demonstrate a range of emergent capabilities, enabling them to process and interpret complex visual and textual information more effectively. However, while grafted VLMs inherit strong linguistic capabilities from their base LLM, they also carry over the propensity to generate ungrounded or fabricated information (Huang et al., 2021; Bang et al., 2023).

### 2.2 HALLUCINATION IN VLMS

Hallucination typically refers to instances in which the generated responses include information that is not present in the visual content (Rohrbach et al., 2018; Biten et al., 2022; Li et al., 2023d). Recent initiatives have aimed to tackle these intricacies, with research focusing on detecting and evaluating object hallucinations in the realm of LVLMs (Wang et al., 2023c; Liu et al., 2023a; Li et al., 2023d; Lovenia et al., 2023; Yin et al., 2024), and methods to reduce them (Liu et al., 2023a; Yin et al., 2024; Wang et al., 2023a). For instance, POPE (Li et al., 2023d) transforms hallucination into a binary classification task to assess the model's ability to recognize whether a particular object is present in the image. Unlike approaches that simply integrate powerful LLMs with in-context or few-shot learning capabilities (Alayrac et al., 2022; Li et al., 2023b), efforts to address hallucinations have primarily focused on incorporating external tools for post-processing. For instance, Woodpecker (Yin et al., 2024) utilizes a five-stage process, but many of these stages rely heavily on auxiliary models, such as multiple LLMs and vision foundation models, making the approach resource-intensive. Additionally, adapting factually augmented reinforcement learning from human feedback (RLHF) (Sun et al., 2023) has emerged as an effective strategy to align model outputs with factual accuracy. However, current strategies (Liu et al., 2024b;a) that involve acquiring additional datasets, performing detailed tuning on initial or new models, or utilizing other pretrained models can be time-intensive, laborious, and computationally demanding.

To address these limitations, several training-free methods have been developed. For instance, Visual Contrastive Decoding (VCD) (Leng et al., 2024) calibrates visual uncertainty by contrasting output distributions generated from original and distorted visual inputs. Similarly, Multi-Modal Mutual Information Decoding (M3ID) (Favero et al., 2024) and Visual Debias Decoding (VDD) (Zhang et al., 2024) enhance the influence of the reference image by comparing probability distributions produced from conditioned and unconditioned inputs. These approaches aim to refine model predictions without requiring additional training. Compared to these methods, our NoLan introduces a fundamentally different, finer-grained assumption. While methods like VCD (Leng et al., 2024) and VDD (Zhang et al., 2024) simplify the problem by assuming a uniform language prior for all tokens, and M3ID assumes that the prior degree is conditioned only on sequence length (Favero et al., 2024), our approach makes a more nuanced and realistic assumption. Specifically, our NoLan posits that each token possesses a distinct language prior. We further propose a simple yet effective KL-based

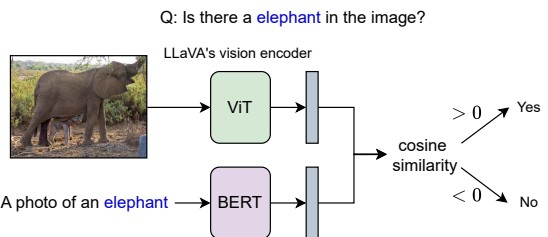

Figure 2: Experimental pipeline to test whether LLaVA's vision encoder can detect the presence of an object in an image.

Table 1: **The Vision encoder can robustly detect object presence in samples.** On the MSCOCO dataset of POPE-*random* (Li et al., 2023d), for samples where LLaVA-1.5 experiences hallucinations, its vision encoder can indeed predict object presence with high accuracy.

| Samples on COCO of POPE-*Random* where LLaVA experiences hallucinations | | | |
|---|---|---|---|
| **Metric** | Accuracy | Precision | Recall | F1 Score |
| **Score** | 83.01 | 83.71 | 98.33 | 90.43 |

method to measure the prior degree of each token. This token-specific and dynamic prior modeling allows our method to more accurately suppress each token's language prior, leading to performance improvements. Thus, our work's novelty lies in this novel assumption and the development of an effective mechanism to model it, which fundamentally distinguishes it from prior work.

## 3 METHOD

### 3.1 PRELIMINARY EXPERIMENTS

LVLMs generally comprise two core components: a vision encoder to gain visual information and a language decoder to generate textual responses. This design raises an important question: are these two components responsible for object hallucinations? In this section, we present a comprehensive analysis to investigate the contributions of both the vision encoder and the language decoder to these hallucinations.

**Vision Encoder.** We aim to investigate whether the vision encoder accurately detects object presence in the failing cases of object hallucinations. To this end, we design a pipeline as shown in Figure 2. Specifically, LLaVA comprises a CLIP vision encoder and a LLaMA (Vicuna) language model, but in this experiment, we use only the CLIP vision encoder. We extract image representation using the CLIP encoder and evaluate whether the representation includes information about a specific object. For this, we transform the text query into "A photo of a [object]" and pass it through CLIP's BERT encoder to obtain a text representation. We then calculate the cosine similarity between CLIP's image and text representations to assess object presence. As shown in Table 1, for samples where LLaVA-1.5 experiences hallucinations on the MSCOCO dataset of POPE (random) (Li et al., 2023d), its vision encoder can predict object presence with high accuracy of 83%. These results lead to our Finding 1: the vision encoder can indeed detect object presence in samples exhibiting object hallucinations.

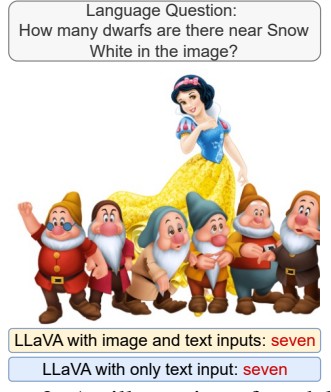

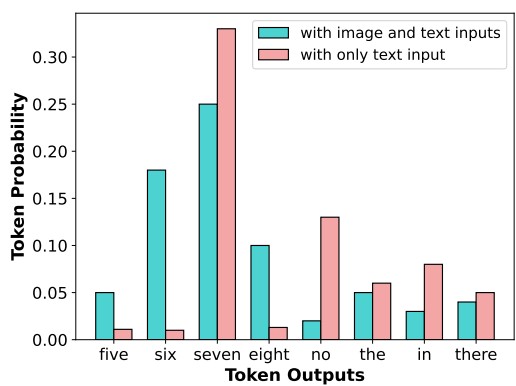

Figure 3: An illustration of model prediction misdirected by the language priors. Given an image depicting six dwarfs in front of Snow White, LLaVA-1.5-13b provides the same token "seven" regardless of whether the image is provided as input or not.

Table 2: **Token probability distribution difference between multimodal and unimodal inputs.** We split MSCOCO with POPE-*random* (Li et al., 2023d) into two subsets according to whether the answers from LLaVA-1.5-7B (Liu et al., 2024a) contain hallucinations or not. Here, $p_m$ and $p_u$ represent the token probability distributions conditioned on multimodal and unimodal (text-only) inputs, respectively. The lower KL Divergence and JS Divergence values in the hallucination subset indicate greater similarity between the two distributions, suggesting that language priors heavily influence the outputs.

| Dataset | $D_{\mathrm{KL}}(p_m\|p_u)$ | $D_{\mathrm{KL}}(p_u\|p_m)$ | $D_{\mathrm{JS}}(p_m,p_u)$ |
|---|---|---|---|
| POPE$_{\mathrm{no-hallucination}}$ | 1.20 | 0.58 | 0.28 |
| POPE$_{\mathrm{hallucination}}$ | 0.46 | 0.28 | 0.11 |

---

**Algorithm 1:** NoLan

**Input:** LVLM $\theta$, textual prompt $x$, image $v$, modulation rate $\alpha$

**Output:** Generated string $y$ conditioned on $x$ and $v$

**Initialization:** $y_0 = \mathrm{BOS}$, $t = 1$

**while** $y_t \neq \mathrm{EOS}$ **do**
  $l_m \leftarrow \mathrm{logit}_\theta(y|v, x, y_{<t})$
  $l_u \leftarrow \mathrm{logit}_\theta(y|x, y_{<t})$
  $\alpha \leftarrow f(l_m, l_u)$; // $f(\cdot)$ can be constant or KL-based
  $l \leftarrow l_m + \alpha(l_m - l_u)$
  $y_t \leftarrow \mathrm{Sampling}(\mathrm{softmax}(l))$
  $t \leftarrow t + 1$
**end**

---

**Language Decoder.** While vision encoders can accurately detect objects, LVLMs - which combine vision encoders with LLaMA-like language decoders - still experience hallucinations. We hypothesize that these hallucinations occur when the output distribution is dominated by language priors embedded in LLMs, as illustrated in Figure 3. To test this hypothesis, we compare output distributions between an LVLM processing image-text inputs and its used LLM processing text-only inputs. Specifically, for LLaVA-1.5-7B (Liu et al., 2024a), we denote:

- $p_m$: Output distribution from LLaVA with image-text inputs.

- $p_u$: Output distribution from LLaVA's language decoder LLaMA with text-only inputs.

We measure the difference between these distributions using KL Divergence and JS Divergence metrics. Using the MSCOCO dataset from POPE-*random* (Li et al., 2023d), we create two subsets based on whether LLaVA-1.5-7B produces hallucinations in its answers. As shown in Table 2, the hallucination subset exhibits significantly smaller divergence between $P_m$ and $P_u$ compared to the no-hallucination subset. This suggests that when hallucinations occur, the model's outputs are more heavily influenced by language priors embedded in LLMs. As shown in Table 2, the distribution difference is prominent in the successful subset, whereas it is minimal in the subset of hallucinated responses. This result confirms that the linguistic priors inherent in the language decoder play a significant role in contributing to hallucinations. Indeed, this model's behavior is not entirely unexpected, as LLMs are fundamentally designed to predict the next word's probability based on extensive textual corpora. When confronted with ambiguous dominant language question stimuli, LVLMs may default to these text-based predictions as a "safety net". While language priors are generally beneficial for contextual understanding and efficient inference, they can introduce biases or assumptions that conflict with the actual visual content. These results lead to our Finding 2: The output distribution of an LVLM is more dominated by its underlying LLM's priors when object hallucinations occur.

### 3.2 NO-LANGUAGE-HALLUCINATION DECODING

While it is commonly believed that hallucinations arise from weak visual signals in the vision module (Guan et al., 2024; Rohrbach et al., 2018; Wang et al., 2023b), our above findings indicate that object hallucinations are primarily driven by language priors. Therefore, in this section, we propose a very simple framework named No-Language-Hallucination Decoding (NoLan), to overcome the influence of language priors on object hallucinations.

Specifically, consider an LVLM parameterized by $\theta$, with visual inputs $v$ and textual inputs $x$. The output $y$ is generated auto-regressively from a probability distribution conditioned on both $v$ and $x$, expressed as:

$$
\begin{aligned}
l_m &= \mathrm{logit}_\theta\left(y_t \mid v, x, y_{<t}\right), \\
y_t &\sim \mathrm{softmax}(l_m), \qquad \text{subject to } y_t \in \mathcal{V},
\end{aligned}
\tag{1}
$$

where $y_t$ represents the token at time step $t$, $y_{<t}$ denotes the sequence of tokens generated up to time $(t-1)$, and $\mathcal{V}$ stands for the vocabulary dictionary.

To obtain the language priors, we feed text only into the model and compute its logits:

$$l_u = \text{logit}_\theta \left( y_t \mid x, y_{<t} \right) \tag{2}$$

Unlike (Leng et al., 2024), the computation of language priors does not rely on distorted visual inputs.

After obtaining regular multimodal logits $l_m$ and language priors $l_u$, the next step is to design the modulation values on output distribution. Inspired by the contrastive decoding in text (Liu et al., 2021; Li et al., 2022b; O'Brien & Lewis, 2023; Shi et al., 2023) and multimodal (Leng et al., 2024; Favero et al., 2024) generation, we compute the difference between $l_m$ and $l_u$ as modulation logits:

$$l_\Delta = \alpha \times (l_m - l_u), \tag{3}$$

where $\alpha$ is a modulation rate that controls the influence of the modulation distribution. Thus, the output probability distribution modulated by $l_\Delta$ can be expressed as:

$$\begin{aligned} p_{\text{nolan}} \left( y \mid v, x \right) &= \text{softmax} \left[ l_m + l_\Delta \right] \\ &= \text{softmax} \left[ \text{logit}_\theta \left( y \mid v, x, y_{<t} \right) \right. \\ &\quad \left. + \alpha (\text{logit}_\theta \left( y \mid v, x, y_{<t} \right) - \text{logit}_\theta \left( y \mid x, y_{<t} \right)) \right], \end{aligned} \tag{4}$$

with $\alpha = 0$ corresponding to standard decoding. Using the adjusted output distribution $p_{\text{nolan}}$, various sampling methods, such as top-p sampling (Holtzman et al., 2019) and beam search (Freitag & Al-Onaizan, 2017), can be applied to generate outputs,

$$y_t \sim p_{\text{nolan}}, \qquad \text{subject to } y_t \in \mathcal{V}. \tag{5}$$

Building on this structure, we propose two versions of NoLan, NoLan-Base and NoLan-Plus, based on the different formulations of modulation term:

NoLan-Base: In this version, $\alpha$ is treated as a fixed hyperparameter, set to 1 by default. Surprisingly, this simple choice already demonstrates impressive performance in our experiments. Then the Equation 4 becomes:

$$\begin{aligned} y_t \sim \text{softmax} & \left[ 2 \times \text{logit}_\theta \left( y_t \mid v, x, y_{<t} \right) \right. \\ & \left. - \text{logit}_\theta \left( y_t \mid x, y_{<t} \right) \right], \\ & \text{subject to } y_t \in \mathcal{V}, \end{aligned} \tag{6}$$

NoLan-Plus: Motivated by Finding 2, as shown in Table 2, which highlights that smaller differences between $l_m$ and $l_u$ are associated with a higher likelihood of hallucinations. To leverage this association, we introduce a self-adjusting mechanism derived from the symmetric KL-divergence, which is expressed as:

$$\gamma = \frac{(D_{\text{KL}}(l_m \| l_u) + D_{\text{KL}}(l_u \| l_m))}{2}, \tag{7}$$

$$\alpha = \beta \times \left( \tanh \left( \frac{1}{\gamma} \right) + 1 \right). \tag{8}$$

The symmetric KL divergence is first inverted and then passed through a tanh function to constrain its range. An additive shift of 1 is subsequently applied to ensure the resulting value lies within the positive domain. To further refine the value range, we introduce a scaling factor $\beta$, which we set to 0.8 based on our ablation experiments in the appendix. This mechanism automatically adjusts the modulation term, effectively suppressing the LLM's priors and enhancing its performance. We refer to this improved version as NoLan-Plus, distinguishing it from the simpler NoLan-Base. A comprehensive theoretical reasoning of this dynamic weighting method can be found in Appendix A.1.

In summary, the NoLan framework introduces two variants: NoLan-Base and NoLan-Plus. Unlike NoLan-Base, which remains static modulation rate throughout generation, NoLan-Plus dynamically adapts to the output distribution, improving both flexibility and effectiveness. We also show the NoLan framework in algorithm 1.

## 4 EXPERIMENTS

In this section, we evaluate NoLan across different LVLMs and tasks to demonstrate its effectiveness.

### 4.1 EXPERIMENTAL SETTINGS

#### 4.1.1 DATASETS & EVALUATION METRICS

**POPE.** The Polling-based Object Probing Evaluation (Li et al., 2023d) (POPE), introduces an efficient method to evaluate object hallucinations. In this benchmark, LVLMs are asked to determine whether a specific object exists in a given image. The POPE benchmark compiles data from three different sources: MSCOCO (Lin et al., 2014), A-OKVQA (Schwenk et al., 2022), and GQA (Hudson & Manning, 2019). The evaluation focuses on four primary metrics: Accuracy, Precision, Recall, and the F1 score.

**MME.** It acts as a comprehensive benchmark for evaluating LVLMs across multiple dimensions (Fu et al., 2023), which encompasses ten subtasks related to perception and four focused on cognition, offering a holistic assessment of multimodal model capabilities. To evaluate hallucinations precisely, we use targeted subsets: existence and count for object-level, and position and color for attribute-level hallucinations. Performance is measured via the composite metric of accuracy and accuracy+ as defined in the official implementation.[1]

**LLaVA-Bench.**[2] This dataset is highly diverse, featuring 24 images paired with 60 questions. It encompasses a wide range of scenarios, including indoor and outdoor scenes, memes, paintings, and sketches, making it an excellent resource for evaluating the capability of LVLMs to handle complex tasks and adapt to diverse domains.

**Other datasets.** Our evaluation also includes benchmarks such as MM-Vet (Yu et al., 2023), MMHal-Bench (Sun et al., 2023), and HallusionBench (Guan et al., 2024), which are detailed in Appendix A.6.

#### 4.1.2 LVLM BASELINES

We evaluate the performance of NoLan across three state-of-the-art LVLMs. To ensure a fair and consistent comparison, our experimental setup aligns with VCD (Leng et al., 2024). Specifically, we integrate NoLan with LLaVA-1.5 (Liu et al., 2024a) and InstructBLIP, both of which use Vicuna 7B as their language decoder (Liu et al., 2023d; Dai et al., 2023), as well as Qwen-VL, which is built on the Qwen 7B backbone (Bai et al., 2023b). More LVLM baselines can be found in Appendix A.4, and the Qwen-VL series is detailed in Appendix A.8.

### 4.2 DECODING BASELINES

One of the decoding methods we compared is direct sampling from the output probability distribution of LVLMs using regular image and text inputs, which we denote as "Regular". A notable training-free method is VCD (Leng et al., 2024), which generates outputs by contrasting distributions from clear and distorted images. Other notable approaches include M3ID (Favero et al., 2024) and VDD (Zhang et al., 2024), which enhance the influence of the reference image while reducing the dominance of language priors. Further contrastive decoding baselines and other attention-based approaches are detailed in appendix A.5 and A.7, respectively.

### 4.3 EXPERIMENTAL RESULTS

**Results on POPE.** Table 3 summarizes the experimental results for POPE under random, popular, and adversarial sampling conditions. A notable highlight is the strong performance of our proposed NoLan approach. NoLan consistently outperforms regular decoding baseline in every evaluated scenario and achieves improvements of up to 8.38 in accuracy and 8.77 in F1 scores across all tested LVLMs. Furthermore, NoLan-Base demonstrates superior performance over VCD (Leng et al., 2024), with improvements of up to 4.56 in accuracy and 2.9 in F1 scores, outperforming VCD in 77.8% of the evaluated cases. NoLan-Plus amplifies this advantage, achieving gains of up to 5.14 in accuracy and

---

[1] https://github.com/BradyFU/Awesome-Multimodal-Large-Language-Models/tree/Evaluation

[2] https://huggingface.co/datasets/liuhaotian/llava-bench-in-the-wild

Table 3: Results on POPE (Li et al., 2023d). *Regular* decoding denotes direct sampling, *VCD* (Leng et al., 2024) indicates sampling from visual contrastive distribution, while methods prefixed with *NoLan* refers to sampling from our proposed contrastive distribution $p_{\text{nolan}}$. The best performances within each setting are bolded. The mean and the standard deviation over 5 runs of POPE.

| Dataset | Model | Decoding | *Random* | | *Popular* | | *Adversarial* | |
|---|---|---|---|---|---|---|---|---|
| | | | Accuracy↑ | F1 Score↑ | Accuracy↑ | F1 Score↑ | Accuracy↑ | F1 Score↑ |
| GQA | LLaVA1.5 | Regular | 83.73(±0.27) | 82.95(±0.28) | 78.17(±0.17) | 78.37(±0.18) | 75.08(±0.33) | 76.06(±0.24) |
| | | VCD | 86.65(±0.45) | 86.99(±0.41) | 80.73(±0.47) | 82.24(±0.35) | 76.09(±0.43) | 78.78(±0.36) |
| | | NoLan-Base (Ours) | 88.35(±0.16) | 87.68(±0.17) | 84.13(±0.30) | 83.94(±0.26) | 80.65(±0.19) | 81.08(±0.21) |
| | | NoLan-Plus (Ours) | **88.53**(±0.10) | **87.84**(±0.12) | **84.62**(±0.33) | **84.35**(±0.21) | **81.23**(±0.17) | **81.56**(±0.19) |
| | Qwen-VL | Regular | 80.97(±0.32) | 79.01(±0.40) | 75.99(±0.35) | 74.84(±0.34) | 75.46(±0.63) | 74.33(±0.71) |
| | | VCD | 85.59(±0.38) | 85.33(±0.38) | 81.83(±0.27) | 82.23(±0.22) | 80.01(±0.27) | 80.75(±0.27) |
| | | NoLan-Base (Ours) | 86.55(±0.22) | 86.13(±0.31) | 82.37(±0.20) | 82.61(±0.19) | 80.23(±0.28) | 80.85(±0.26) |
| | | NoLan-Plus (Ours) | **87.27**(±0.22) | **87.04**(±0.17) | **83.20**(±0.24) | **83.61**(±0.15) | **80.25**(±0.31) | **81.06**(±0.17) |
| | InstructBLIP | Regular | 79.65(±0.24) | 80.56(±0.19) | 73.87(±0.58) | 76.42(±0.52) | 70.56(±0.53) | 74.12(±0.58) |
| | | VCD | 83.69(±0.11) | 84.16(±0.01) | 78.57(±0.14) | 80.17(±0.16) | 75.08(±0.13) | 77.53(±0.08) |
| | | NoLan-Base (Ours) | 85.62(±0.28) | 85.02(±0.18) | 79.61(±0.02) | 80.00(±0.21) | 77.00(±0.15) | 77.97(±0.13) |
| | | NoLan-Plus (Ours) | **86.15**(±0.11) | **85.27**(±0.19) | **81.12**(±0.21) | **80.99**(±0.17) | **78.13**(±0.12) | **78.43**(±0.10) |
| A-OKVQA | LLaVA1.5 | Regular | 83.45(±0.48) | 82.56(±0.50) | 79.90(±0.33) | 79.59(±0.37) | 74.04(±0.34) | 75.15(±0.23) |
| | | VCD | 86.15(±0.23) | 86.34(±0.21) | 81.85(±0.44) | 82.82(±0.36) | 74.97(±0.39) | 77.73(±0.29) |
| | | NoLan-Base (Ours) | 87.83(±0.16) | 87.21(±0.19) | 85.41(±0.42) | 85.00(±0.42) | 79.21(±0.20) | 79.90(±0.17) |
| | | NoLan-Plus (Ours) | **88.04**(±0.14) | **87.32**(±0.16) | **85.85**(±0.20) | **85.36**(±0.19) | **79.61**(±0.17) | **80.19**(±0.16) |
| | Qwen-VL | Regular | 86.67(±0.48) | 85.59(±0.53) | 85.56(±0.35) | 84.63(±0.42) | 79.57(±0.31) | 79.50(±0.38) |
| | | VCD | 89.22(±0.14) | 89.01(±0.14) | 87.85(±0.30) | 87.81(±0.31) | 81.27(±0.09) | **82.38**(±0.10) |
| | | NoLan-Base (Ours) | 89.17(±0.28) | 88.80(±0.33) | 87.42(±0.29) | 87.10(±0.28) | 81.10(±0.21) | 81.91(±0.27) |
| | | NoLan-Plus (Ours) | **89.40**(±0.20) | **89.02**(±0.13) | **88.00**(±0.16) | **87.83**(±0.24) | 81.20(±0.19) | 82.06(±0.15) |
| | InstructBLIP | Regular | 80.91(±0.34) | 81.86(±0.32) | 76.19(±0.80) | 78.17(±0.73) | 70.71(±0.76) | 75.56(±0.57) |
| | | VCD | 84.11(±0.27) | 84.56(±0.28) | 79.78(±0.47) | 81.15(±0.42) | 74.33(±0.67) | 77.19(±0.47) |
| | | NoLan-Base (Ours) | 87.87(±0.37) | 87.46(±0.32) | 83.60(±0.43) | 83.76(±0.31) | 77.33(±0.45) | 78.79(±0.47) |
| | | NoLan-Plus (Ours) | **88.20**(±0.33) | **87.55**(±0.21) | **84.57**(±0.42) | **84.32**(±0.36) | **78.43**(±0.22) | **79.24**(±0.27) |
| MSCOCO | LLaVA1.5 | Regular | 83.29(±0.35) | 81.33(±0.41) | 81.88(±0.48) | 80.06(±0.05) | 78.96(±0.42) | 77.57(±0.57) |
| | | VCD | **87.73**(±0.40) | **87.16**(±0.41) | 85.38(±0.38) | 85.06(±0.37) | 80.88(±0.33) | 81.13(±0.34) |
| | | NoLan-Base (Ours) | 86.73(±0.15) | 85.15(±0.20) | 85.63(±0.17) | 84.12(±0.21) | 83.22(±0.17) | 81.93(±0.22) |
| | | NoLan-Plus (Ours) | 87.11(±0.13) | 86.60(±0.16) | **85.81**(±0.13) | **85.17**(±0.17) | **83.83**(±0.17) | **82.58**(±0.16) |
| | Qwen-VL | Regular | 84.73(±0.36) | 82.67(±0.41) | 84.13(±0.18) | 82.06(±0.23) | 82.26(±0.30) | 80.37(±0.37) |
| | | VCD | **88.63**(±0.10) | **87.81**(±0.11) | 87.12(±0.07) | 86.40(±0.09) | 84.26(±0.39) | 83.90(±0.39) |
| | | NoLan-Base (Ours) | 88.30(±0.19) | 87.22(±0.20) | 86.83(±0.27) | 85.70(±0.25) | 84.91(±0.31) | 84.01(±0.33) |
| | | NoLan-Plus (Ours) | 88.10(±0.11) | 87.00(±0.10) | **87.43**(±0.29) | **86.43**(±0.22) | **84.93**(±0.18) | **84.07**(±0.17) |
| | InstructBLIP | Regular | 80.71(±0.73) | 80.41(±0.80) | 78.22(±0.84) | 78.36(±0.76) | 75.84(±0.45) | 76.59(±0.40) |
| | | VCD | 84.53(±0.38) | 83.68(±0.40) | 81.47(±0.42) | 81.07(±0.39) | 79.56(±0.41) | 79.52(±0.38) |
| | | NoLan-Base (Ours) | **86.07**(±0.41) | **84.45**(±0.36) | 83.97(±0.33) | 82.43(±0.28) | 81.97(±0.48) | 80.75(±0.44) |
| | | NoLan-Plus (Ours) | 85.67(±0.33) | 83.81(±0.31) | **84.00**(±0.26) | **82.49**(±0.30) | **82.37**(±0.19) | **80.81**(±0.23) |

Table 4: Results of accuracy on MSCOCO of POPE using the setting in M3ID (Favero et al., 2024). We follow M3ID using its template: "Is a ⟨object⟩ present in the image?" for a fair comparison.

| Decoding | MSCOCO of POPE | | | |
|---|---|---|---|---|
| | Random ↑ | Popular ↑ | Adversarial ↑ | All ↑ |
| *LLaVA-1.5-7B* | | | | |
| Regular | 74.8 | 61.8 | 58.1 | 64.9 |
| M3ID | 76.0 | 69.3 | 65.8 | 70.3 |
| NoLan-Base (Ours) | 87.8 | 86.3 | 82.7 | 85.6 |
| NoLan-Plus (Ours) | **88.8** | **87.5** | **83.7** | **86.7** |
| *LLaVA-1.5-13B* | | | | |
| Regular | 67.9 | 63.8 | 59.8 | 63.8 |
| M3ID | 84.3 | 77.0 | 71.3 | 77.5 |
| NoLan-Base (Ours) | 88.0 | 86.8 | 84.0 | 86.3 |
| NoLan-Plus (Ours) | **89.2** | **88.3** | **85.2** | **87.6** |

Table 5: Results of accuracy on MSCOCO of POPE using the setting in VDD (Zhang et al., 2024). We follow the decoding format and evaluation settings in VDD to ensure a fair comparison.

| Decoding | MSCOCO of POPE | | | |
|---|---|---|---|---|
| | Random ↑ | Popular ↑ | Adversarial ↑ | All ↑ |
| *LLaVA-1.5-7B* | | | | |
| Regular | 83.29 | 81.88 | 78.96 | 81.37 |
| VDD | 87.07 | **85.87** | 83.52 | 85.49 |
| NoLan-Base (Ours) | 86.50 | 85.13 | 83.00 | 84.89 |
| NoLan-Plus (Ours) | **87.10** | 85.83 | **83.63** | **85.52** |
| *LLaVA-1.5-13B* | | | | |
| Regular | 83.31 | 82.47 | 80.00 | 81.92 |
| VDD | 86.88 | 86.08 | 84.34 | 85.77 |
| NoLan-Base (Ours) | 87.37 | 86.23 | 83.87 | 85.82 |
| NoLan-Plus (Ours) | **88.70** | **87.40** | **84.90** | **87.00** |

3.17 in F1 scores, surpassing VCD in 88.9% of the experiments. With the template in M3ID (Favero et al., 2024), NoLan significantly suppresses M3ID in accuracy, achieving improvements of up to 18.2 and 13.9, with an average increase of 16.4 and 10.1 on the 7B and 13B models, respectively. Additionally, to ensure a fair comparison, when using the same settings as VDD (Zhang et al., 2024), NoLan-Plus still outperforms VDD on both the 7B and 13B models. This underscores NoLan's effectiveness in mitigating object hallucinations in LVLMs, emphasizing that object hallucinations are predominantly driven by language priors in language models. By weakening these priors at a more fine-grained level, the model's performance can be significantly improved.

**Results on MME hallucination subset.** The evaluation on the MME subset extends beyond POPE by addressing both object-level and attribute-level hallucinations. As presented in Table 6, the implementation of NoLan consistently improves performance across all models in resolving attribute-level hallucinations. Notably, NoLan-Plus outperforms both the regular baseline, VCD and VDD on the majority of subsets, further underscoring its effectiveness. In terms of object-level hallucinations, both variants of NoLan show a clear positive impact on the *Existence* metric, significantly enhancing overall performance. These improvements emphasize NoLan's strength in mitigating object hallucinations across diverse scenarios.

**Case study on LLaVA-Bench.** Figure 4 illustrates two case studies that demonstrate the effectiveness of NoLan-Plus in mitigating object hallucinations. In the cases presented, objects like "*suitcase*" and "*truck*" which are commonly associated with the ground truth object "*taxi*", erroneously appear as hallucinations in the generated output. In contrast, the application of NoLan-Plus significantly reduces these hallucinations while preserving the consistency and richness of the generated text.

Table 6: Results on the hallucination subset of MME (Fu et al., 2023). Regular decoding denotes direct sampling, VCD (Leng et al., 2024) indicates sampling from visual contrastive distribution, and VDD (Zhang et al., 2024) expresses visual debias decoding. In contrast, methods prefixed with NoLan refer to sampling from our proposed contrastive distribution $p_{\mathrm{nolan}}$. The best performances within each setting are **bolded**.

| Model | Decoding | Object-level | | Attribute-level | | Total Scores↑ |
|---|---|---|---|---|---|---|
| | | Existence↑ | Count↑ | Position↑ | Color↑ | |
| LLaVA1.5 | Regular | 175.67 | 124.67 | 114.00 | 151.00 | 565.33 |
| | VCD | 184.66 | 138.33 | 128.67 | 153.00 | 604.66 |
| | VDD | **190.00** | 143.30 | **145.00** | 165.00 | 643.29 |
| | NoLan-Base (Ours) | **190.00** | 145.00 | 138.33 | 155.00 | 628.33 |
| | NoLan-Plus (Ours) | **190.00** | **151.67** | 143.33 | **175.00** | **660.00** |
| Qwen-VL | Regular | 155.00 | 127.67 | 131.67 | 173.00 | 587.33 |
| | VCD | 156.00 | 131.00 | 128.00 | 181.67 | 596.67 |
| | VDD | 165.00 | **145.00** | 148.30 | **190.00** | 643.29 |
| | NoLan-Base (Ours) | 160.00 | 135.00 | 133.33 | **190.00** | 618.33 |
| | NoLan-Plus (Ours) | **185.00** | **145.00** | 138.33 | 180.00 | **648.33** |
| InstructBLIP | Regular | 141.00 | 75.33 | 66.67 | 97.33 | 380.33 |
| | VCD | 168.33 | **92.33** | 64.00 | 123.00 | 447.67 |
| | NoLan-Base (Ours) | 175.00 | 61.67 | 68.33 | 118.33 | 423.33 |
| | NoLan-Plus (Ours) | **180.00** | 65.00 | **76.67** | **138.33** | **460.00** |

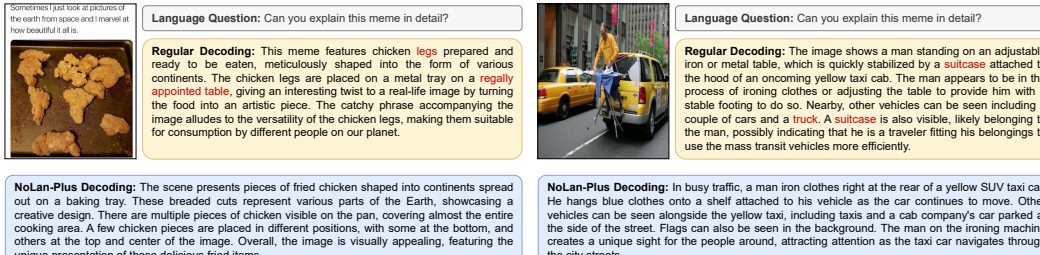

Figure 4: Illustration of hallucination mitigation by our proposed NoLan-Plus with two samples from LLaVA-Bench. Hallucinated objects from LVLM's regular decoding are highlighted in red.

This showcases NoLan-Plus's ability to produce outputs that are more aligned with the visual input without sacrificing informativeness. Due to space constraints, additional case studies are included in Appendix A.11 for further reference.

## 5 CONCLUSION AND DISCUSSION

In this paper, we tackle the critical challenge of object hallucinations in LVLMs. We begin by analyzing the roles of the vision encoder and language decoder in contributing to these hallucinations. Our experiments reveal that in hallucination cases, vision encoders effectively detect objects; however, the output distribution is heavily influenced by the priors of the language decoder. Based on this insight, we propose No-Language-Hallucination Decoding (NoLan), a simple, training-free framework to overcome language priors. It leverages contrastive distributions from multimodal and text-only inputs, to refine the model's outputs, without relying on external tools. This structure introduces two method variants: NoLan-Base and NoLan-Plus. While NoLan-Base maintains a constant configuration throughout generation, NoLan-Plus dynamically adjusts to the output distribution, offering greater flexibility and improved effectiveness. NoLan operates during inference and can be seamlessly integrated with any pre-trained autoregressive LVLMs. This design makes NoLan a cost-effective and flexible solution for improving vision-language grounding. Extensive experiments conducted across diverse benchmarks and architectures of LVLMs validate NoLan's effectiveness in mitigating object hallucinations.

**Limitation and future work.** One limitation of NoLan is its need for two forward passes during inference, although the second pass is more resource-efficient than that of VCD and VDD. To reduce the inference time, an alternative approach could involve processing two sets of batched queries concurrently, with one set containing masked visual tokens. Additionally, our study focuses solely on LVLMs designed for image and text processing. Expanding the NoLan framework to support a wider range of LVLMs represents a promising direction for future development.

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

## A  APPENDIX

### A.1  THEORETICAL PROOF OF NOLAN-PLUS

The core mechanism of NoLan-Plus is a dynamic weighting method that uses KL divergence to measure the difference between multimodal and text-only output distributions. In addition to the empirical justification and experimental research for using KL divergence, we conduct the following theoretical analysis.

**Definition of Visual Object Hallucination.**

Visual Object Hallucination is defined as the case where the model's response $y$ is almost independent of the visual input $v$ given a textual prompt $x$. Formally, this dependency is quantified using the conditional mutual information:

$$I(y; v \mid x). \tag{9}$$

A lower mutual information implies stronger hallucination. As an extreme case, if $y$ and $v$ are independent, then the response is generated without reference to the visual input, and

$$I(y; v \mid x) = 0. \tag{10}$$

**Connecting Conditional Mutual Information to KL Divergence.**

For a fixed prompt $x$, define

$$p_m(y) = P_\theta(y \mid v, x), \quad p_u(y) = P_\theta(y \mid x), \tag{11}$$

where $p_m$ is the output distribution conditioned on both image and text, and $p_u$ is conditioned on text only.

By the standard identity between conditional mutual information and KL divergence, we have:

$$I(y; v \mid x) = \mathbb{E}_{v|x}\Big[D_{\mathrm{KL}}\big(p_m \,\|\, p_u\big)\Big]. \tag{12}$$

*Proof.* Starting from the definition of conditional mutual information:

$$I(y; v \mid x) = \mathbb{E}_{v,y|x}\left[\log \frac{P(y, v \mid x)}{P(y \mid x)P(v \mid x)}\right] \tag{13}$$

$$= \mathbb{E}_{v,y|x}\left[\log \frac{P(y \mid v, x)}{P(y \mid x)}\right] \tag{14}$$

$$= \mathbb{E}_{v|x}\left[\sum_y P(y \mid v, x) \log \frac{P(y \mid v, x)}{P(y \mid x)}\right] \tag{15}$$

$$= \mathbb{E}_{v|x}\left[D_{\mathrm{KL}}\big(P(Y \mid v, x) \,\|\, P(Y \mid x)\big)\right] \tag{16}$$

$$= \mathbb{E}_{v|x}\left[D_{\mathrm{KL}}\big(p_m \,\|\, p_u\big)\right]. \tag{17}$$

$$\square$$

Thus, a lower KL divergence $D_{\mathrm{KL}}(p_m \,\|\, p_u)$ indicates a lower mutual information between the visual input and the response, leading to stronger hallucination.

### A.2  UNCERTAINTY ANALYSIS AND LANGUAGE PRIOR SUPPRESSION

A core motivation of our approach is to mitigate the influence of language priors in vision-language models. To this end, we adopt a contrastive decoding strategy that reshapes the output distribution without additional training. While this training-free formulation effectively suppresses dominant linguistic priors, it may also introduce instability into the decoding process: by altering token probabilities post hoc, it can distort the relative ranking of non-target tokens and occasionally amplify spurious modes in the distribution.

Table 7: Entropy-based uncertainty evaluation across four benchmarks: POPE, MME, MM-Vet, and LLaVA-Bench. Results indicate that NoLan achieves the lowest uncertainty in all settings.

| Decoding | POPE ↓ | MME ↓ | MM-Vet ↓ | LLaVA-Bench ↓ |
|---|---|---|---|---|
| Regular | 0.6484 | 2.9863 | 1.5889 | 4.1875 |
| Text-only | 0.6948 | 3.7051 | 2.0020 | 3.5508 |
| VCD | 0.4646 | 0.8632 | 0.6854 | 2.1153 |
| NoLan-Base | 0.4692 | 0.9106 | 0.6040 | 0.8188 |
| NoLan-Plus | **0.3786** | **0.7439** | **0.4423** | **0.7931** |

To evaluate these effects more explicitly, we analyze the LLaVA-1.5's predictive uncertainty using entropy over output distributions. Lower entropy indicates greater confidence and better calibration.

As shown in Table 7, we report entropy across four diverse benchmarks: POPE, MME, MM-Vet, and LLaVA-Bench. Compared to regular decoding and the text-only baseline, both NoLan variants yield substantially lower entropy in all settings. In particular, NoLan-Plus achieves the lowest uncertainty, suggesting that our method not only suppresses linguistic bias but also maintains overall distributional stability in most cases.

## A.3 CORRELATION STUDY BETWEEN HALLUCINATION AND TOKEN POSITION

Table 8: Token-wise KL values indicating hallucination.

| Token Pos | 0 | 1 | 2 | 3 | 4 | 5 | 6 | 7 | 8 | 9 | 10 | 11 | 12 |
|---|---|---|---|---|---|---|---|---|---|---|---|---|---|
| KL Value | 1.36 | 0.53 | 0.97 | 0.50 | 0.57 | 0.76 | 0.63 | 0.66 | 0.47 | 0.46 | 0.36 | 0.42 | 0.42 |

Recent studies have highlighted that the position of a generated token within a sequence can be a significant factor in the emergence of model hallucinations. For instance, M3ID (Favero et al., 2024), demonstrated that as a model generates more tokens, its reliance on the initial visual prompt decreases, leading to an increase in hallucinations. Based on this finding, we conducted a study to quantitatively measure this correlation between token positions and the degree of hallucination. Specifically, we used LLaVA-1.5 7B to evaluate performance across the entire LLaVA-Bench. Following the method in our preliminary experiments, we calculated the mean KL divergence across all samples at each token position to estimate the likelihood of hallucination. Here, token position refers to the index assigned to each generated token after the model receives the input image and text. The first generated token is assigned position 0, and subsequent tokens are indexed sequentially based on their order in the output sequence. The results are shown in Table 8. The observation is generally in line with the findings of M3ID. Overall, the experimental results show that the farther a token is from the beginning of the sequence, the more similar the distributions of the two forward passes become, and the greater the likelihood of hallucination. However, some samples deviate from this trend, possibly because the output tokens are not strictly object-related but also include many non-object terms. As a result, the values do not consistently vary with token position.

## A.4 ABLATION STUDY

We use the same LVLM baselines for ablation studies.

**Modulation rate.** The parameter $\alpha$ governs the amplification of the modulation distribution generated from multimodal and unimodal inputs, as defined in Equation 3. We adjust $\alpha$ to examine its impact on NoLan-Base and identify the optimal value for performance. As shown in Table 9, $\alpha = 1$ yields the best or second-best performance, so we set it as the default. Similarly, the parameter $\beta$ in NoLan-Plus, which controls the boundary of the auto-adjusting modulation rate, achieves optimal or suboptimal results at $\beta = 0.8$, as demonstrated in Table 10, making it our default setting.

**Logit components.** The logits in NoLan consist of regular multimodal logits $l_m$, derived from image and text inputs, and unimodal logits $l_u$, derived from text-only inputs. As shown in Table 11, using only $l_m$ or $l_u$ results in a significant performance drop, highlighting the critical role of each logit component in NoLan and the effectiveness of its utilization mechanism. This finding further supports our hypothesis that object hallucinations predominantly originate from linguistic priors.

Table 9: **Sensitivity to modulation rate** $\alpha$**.** In NoLan-Base, $\alpha$ is manually set to regulate the influence of the modulation distribution, defined in Equation 3. When $\alpha = 0$, NoLan-Base reverts to standard decoding.

| | | POPE | | | | | MME | | | | MM-Vet |
|---|---|---|---|---|---|---|---|---|---|---|---|
| $\alpha$ | Model | Accuracy ↑ | Precision | Recall | F1 Score ↑ | MME-Hallu ↑ | Existence ↑ | Count ↑ | Position ↑ | Color ↑ | total ↑ |
| 0.0 | LLaVA-1.5-7B | 83.29 | 92.13 | 72.80 | 81.33 | 565.33 | 175.67 | 124.67 | 114.00 | 151.00 | 31.1 |
| 1.0 | LLaVA-1.5-7B | **86.50** | 96.68 | 75.60 | **84.85** | **628.33** | 190.00 | 145.00 | 138.33 | 155.00 | **33.0** |
| 2.0 | LLaVA-1.5-7B | 86.27 | 96.66 | 75.13 | 84.55 | 620.00 | 190.00 | 143.33 | 138.33 | 148.33 | 32.5 |
| 3.0 | LLaVA-1.5-7B | 85.97 | 96.63 | 74.53 | 84.16 | 588.33 | 180.00 | 138.33 | 121.67 | 148.33 | 31.8 |
| 0.0 | LLaVA-1.5-13B | 84.35 | 93.22 | 74.04 | 82.60 | 616.67 | 185.00 | 136.67 | 131.67 | 163.33 | 36.1 |
| 1.0 | LLaVA-1.5-13B | **87.37** | 95.61 | 78.33 | **86.11** | **636.67** | 190.00 | 165.00 | 128.33 | 153.33 | **37.6** |
| 2.0 | LLaVA-1.5-13B | 87.23 | 97.03 | 76.56 | 85.62 | 620.00 | 190.00 | 145.00 | 138.33 | 155.00 | 36.8 |
| 3.0 | LLaVA-1.5-13B | 87.12 | 96.94 | 75.91 | 85.33 | 631.67 | 190.00 | 150.00 | 133.33 | 158.33 | 36.7 |
| 0.0 | Qwen-VL | 84.73 | 95.61 | 72.81 | 82.67 | 587.33 | 155.00 | 127.67 | 131.67 | 173.00 | 33.7 |
| 1.0 | Qwen-VL | **88.30** | 96.07 | 79.87 | **87.22** | **618.33** | 160.00 | 135.00 | 133.33 | 190.00 | 34.5 |
| 2.0 | Qwen-VL | 87.93 | 95.02 | 80.07 | 86.90 | 613.33 | 165.00 | 135.00 | 133.33 | 180.00 | **34.7** |
| 3.0 | Qwen-VL | 87.87 | 94.72 | 80.20 | 86.86 | 613.33 | 170.00 | 135.00 | 133.33 | 175.00 | 34.0 |
| 0.0 | InstructBLIP-7B | 80.71 | 81.67 | 79.19 | 80.41 | 380.33 | 141.00 | 75.33 | 66.67 | 97.33 | 25.2 |
| 1.0 | InstructBLIP-7B | **85.57** | 96.76 | 73.60 | **83.60** | **423.33** | 175.00 | 61.67 | 68.33 | 118.33 | **25.7** |
| 2.0 | InstructBLIP-7B | 84.20 | 97.32 | 70.33 | 81.66 | 413.33 | 180.00 | 50.00 | 58.33 | 125.00 | 25.4 |
| 3.0 | InstructBLIP-7B | 83.73 | 98.01 | 68.87 | 80.89 | 406.67 | 165.00 | 55.00 | 63.33 | 123.33 | 25.5 |
| 0.0 | InstructBLIP-13B | 81.92 | 83.13 | 80.44 | 81.75 | 440.00 | 160.00 | 60.00 | 66.67 | 153.33 | 21.2 |
| 1.0 | InstructBLIP-13B | **86.70** | 97.21 | 75.49 | **84.80** | **465.00** | 180.00 | 65.00 | 76.67 | 143.33 | 25.4 |
| 2.0 | InstructBLIP-13B | 85.43 | 97.72 | 72.15 | 83.03 | 460.00 | 180.00 | 60.00 | 76.67 | 143.33 | **25.5** |
| 3.0 | InstructBLIP-13B | 84.81 | 98.25 | 70.90 | 82.20 | 450.00 | 180.00 | 60.00 | 66.67 | 143.33 | **25.5** |

Table 10: **Sensitivity to modulation rate** $\beta$**.** In NoLan-Plus, $\beta$ is manually set to regulate the influence of the modulation distribution, defined in Equation 8. When $\beta = 0$, NoLan-Plus reverts to standard decoding.

| | | POPE | | | | | MME | | | | MM-Vet ↑ |
|---|---|---|---|---|---|---|---|---|---|---|---|
| $\beta$ | Model | Accuracy ↑ | Precision | Recall | F1 Score ↑ | MME-Hallu ↑ | Existence ↑ | Count ↑ | Position ↑ | Color ↑ | total ↑ |
| 0.0 | LLaVA-1.5-7B | 83.29 | 92.13 | 72.80 | 81.33 | 565.33 | 175.67 | 124.67 | 114.00 | 151.00 | 31.1 |
| 0.2 | LLaVA-1.5-7B | 86.37 | 96.66 | 75.33 | 84.68 | 588.33 | 180.00 | 138.33 | 121.67 | 148.33 | 30.5 |
| 0.4 | LLaVA-1.5-7B | 86.33 | 96.58 | 75.33 | 84.64 | 626.67 | 190.00 | 155.00 | 128.33 | 153.33 | 32.8 |
| 0.6 | LLaVA-1.5-7B | 86.60 | 96.76 | 75.73 | 84.97 | 645.00 | 190.00 | 143.33 | 143.33 | 168.33 | 32.5 |
| 0.8 | LLaVA-1.5-7B | **87.00** | 97.27 | 76.13 | **85.42** | **660.00** | 190.00 | 151.67 | 143.33 | 175.00 | **33.3** |
| 1.0 | LLaVA-1.5-7B | 86.83 | 97.10 | 75.93 | 85.22 | 631.67 | 190.00 | 150.00 | 133.33 | 158.33 | 32.7 |
| 0.0 | LLaVA-1.5-13B | 83.31 | 91.46 | 73.48 | 81.49 | 616.67 | 185.00 | 136.67 | 131.67 | 163.33 | 36.1 |
| 0.2 | LLaVA-1.5-13B | 85.60 | 93.84 | 76.20 | 84.11 | 620.00 | 190.00 | 143.33 | 138.33 | 148.33 | 36.8 |
| 0.4 | LLaVA-1.5-13B | 86.30 | 97.14 | 74.80 | 84.52 | 646.67 | 190.00 | 145.00 | 138.33 | 173.33 | 36.5 |
| 0.6 | LLaVA-1.5-13B | 87.03 | 97.28 | 76.20 | 85.46 | 630.00 | 190.00 | 148.33 | 133.33 | 158.33 | 36.7 |
| 0.8 | LLaVA-1.5-13B | **88.70** | 96.03 | 80.73 | **87.72** | **656.67** | 190.00 | 145.00 | 143.33 | 178.33 | **38.3** |
| 1.0 | LLaVA-1.5-13B | 86.97 | 97.11 | 76.20 | 85.39 | 646.67 | 190.00 | 145.00 | 138.33 | 173.33 | 35.8 |
| 0.0 | Qwen-VL | 84.73 | 95.61 | 72.81 | 82.67 | 587.33 | 155.00 | 127.67 | 131.67 | 173.00 | 33.7 |
| 0.2 | Qwen-VL | 85.03 | 89.54 | 79.33 | 84.13 | 626.67 | 170.00 | 138.33 | 138.33 | 180.00 | 34.0 |
| 0.4 | Qwen-VL | 87.13 | 94.07 | 79.27 | 86.03 | 613.33 | 165.00 | 135.00 | 133.33 | 180.00 | 34.3 |
| 0.6 | Qwen-VL | 87.90 | 95.81 | 79.27 | 86.76 | 618.33 | 170.00 | 135.00 | 138.33 | 175.00 | 33.6 |
| 0.8 | Qwen-VL | **88.10** | 95.83 | 79.67 | **87.00** | **648.33** | 185.00 | 145.00 | 138.33 | 180.00 | **35.2** |
| 1.0 | Qwen-VL | 87.73 | 96.32 | 78.47 | 86.48 | 628.33 | 185.00 | 135.00 | 133.33 | 175.00 | 34.8 |
| 0.0 | InstructBLIP-7B | 80.71 | 81.67 | 79.19 | 80.41 | 380.33 | 141.00 | 75.33 | 66.67 | 97.33 | 25.2 |
| 0.2 | InstructBLIP-7B | 83.53 | 87.04 | 78.80 | 82.72 | 408.33 | 175.00 | 50.00 | 58.33 | 125.00 | 24.5 |
| 0.4 | InstructBLIP-7B | 83.73 | 92.03 | 73.87 | 81.95 | 413.33 | 180.00 | 50.00 | 58.33 | 125.00 | 25.0 |
| 0.6 | InstructBLIP-7B | **85.80** | 91.63 | 78.80 | **84.73** | 396.67 | 170.00 | 50.00 | 58.33 | 118.33 | 25.5 |
| 0.8 | InstructBLIP-7B | 85.67 | 96.28 | 74.20 | 83.81 | **460.00** | 180.00 | 65.00 | 76.67 | 138.33 | 27.0 |
| 1.0 | InstructBLIP-7B | 85.57 | 96.43 | 73.87 | 83.65 | 406.67 | 165.00 | 55.00 | 63.33 | 123.33 | **27.3** |
| 0.0 | InstructBLIP-13B | 82.36 | 86.93 | 76.19 | 81.20 | 440.00 | 160.00 | 60.00 | 66.67 | 153.33 | 21.2 |
| 0.2 | InstructBLIP-13B | 83.43 | 92.11 | 73.13 | 81.53 | 441.67 | 180.00 | 60.00 | 58.33 | 143.33 | 24.3 |
| 0.4 | InstructBLIP-13B | 85.13 | 96.72 | 72.73 | 83.03 | 440.00 | 178.33 | 66.67 | 60.00 | 135.00 | 26.1 |
| 0.6 | InstructBLIP-13B | 88.07 | 93.92 | 81.40 | 87.21 | 488.33 | 180.00 | 88.33 | 76.67 | 143.33 | 25.8 |
| 0.8 | InstructBLIP-13B | **88.90** | 94.92 | 82.20 | **88.10** | **503.33** | 180.00 | 88.33 | 91.67 | 143.33 | **26.7** |
| 1.0 | InstructBLIP-13B | 85.57 | 88.24 | 82.07 | 85.04 | 436.67 | 180.00 | 60.00 | 53.33 | 143.33 | 25.5 |

**LVLMs' model size.** Our evaluation extends to the larger 13B variants of the LLaVA-1.5 (Liu et al., 2024a) and InstructBLIP (Dai et al., 2023), examining the scalability of our proposed NoLan across different LVLM sizes and architectures. Table 12 illustrates that the 7B and 13B variants of LLaVA-1.5 and InstructBLIP deliver comparable performances across POPE settings (e.g., F1 scores of 78.36 and 78.35 for InstructBLIP 7B and 13B in the *Popular* setting), indicating that increasing model parameters alone does not inherently resolve hallucination issues. Notably, NoLan uniformly exceeds the regular method in every evaluated case. Its improvements are particularly pronounced with larger models. These results highlight NoLan's effectiveness and robustness across varying model scales and architectures.

**Variations of NoLan-Plus.** We incorporate the *tanh* function into the Kullback-Leibler Divergence-based function of NoLan-Plus, as defined in Equation 8. Given the similar mathematical properties of the *sigmoid* and *tanh* functions, we conduct an in-depth analysis of both to evaluate their efficiency and generalization potential. As illustrated in Table 13, *tanh* consistently outperforms *sigmoid*

Table 11: **Ablation studies for components of decoding logits and model sizes.** For LLaVA-1.5 on MSCOCO of POPE-*random*, the performance drops significantly with only multimodal or text-only logits. Additionally, the performance gap between NoLan and other decoding methods increases as the model size grows.

| Decoding logits | | Accuracy ↑ | Precision | Recall | F1 Score ↑ |
|---|---|---|---|---|---|
| | | MSCOCO of POPE-*random* | | | |
| | | *LLaVA-1.5-7B* | | | |
| $l_m + \alpha(l_m - l_u)$ | NoLan-Plus | **87.00** | **97.27** | **76.13** | **85.42** |
| | NoLan-Base | 86.50 | 96.68 | 75.60 | 84.85 |
| $l_m$ | multimodal | 83.29 | 92.13 | 72.80 | 81.33 |
| $l_u$ | text-only | 47.57 | 47.45 | 45.33 | 46.37 |
| | | *LLaVA-1.5-13B* | | | |
| $l_m + \alpha(l_m - l_u)$ | NoLan-Plus | **88.70** | **96.03** | **80.73** | **87.72** |
| | NoLan-Base | 87.37 | 95.61 | 78.33 | 86.11 |
| $l_m$ | multimodal | 83.31 | 91.46 | 73.48 | 81.49 |
| $l_u$ | text-only | 49.43 | 49.13 | 32.00 | 38.76 |

Table 12: **Ablation study for LVLMs' model sizes on MSCOCO of POPE.** Scaling up LVLM model sizes does not significantly mitigate object hallucinations. In contrast, NoLan consistently enhances model performance.

| Dataset | POPE | Model | Decoding | Accuracy ↑ | Precision | Recall | F1 Score ↑ |
|---|---|---|---|---|---|---|---|
| MSCOCO | Random | LLaVA1.5-7B | Regular | 83.29 | 92.13 | 72.80 | 81.33 |
| | | | NoLan-Base | 86.50 | 96.68 | 75.60 | 84.85 |
| | | | NoLan-Plus | 87.00 | 97.27 | 76.13 | 85.42 |
| | | LLaVA1.5-13B | Regular | 83.31 | 91.46 | 73.48 | 81.49 |
| | | | NoLan-Base | 87.37 | 95.61 | 78.33 | 86.11 |
| | | | NoLan-Plus | **88.70** | 96.03 | 80.73 | **87.72** |
| | | InstructBLIP-7B | Regular | 80.71 | 81.67 | 79.19 | 80.41 |
| | | | NoLan-Base | 86.07 | 95.54 | 75.67 | 84.45 |
| | | | NoLan-Plus | 85.67 | 96.28 | 74.20 | 83.81 |
| | | InstructBLIP-13B | Regular | 82.36 | 86.93 | 76.19 | 81.20 |
| | | | NoLan-Base | 88.63 | 93.92 | 81.40 | 87.83 |
| | | | NoLan-Plus | **88.90** | 94.92 | 82.20 | **88.10** |
| | Popular | LLaVA1.5-7B | Regular | 81.88 | 88.93 | 72.80 | 80.06 |
| | | | NoLan-Base | 85.13 | 93.41 | 75.60 | 83.57 |
| | | | NoLan-Plus | 85.60 | 93.91 | 76.13 | 84.09 |
| | | LLaVA1.5-13B | Regular | 82.47 | 89.55 | 73.53 | 80.75 |
| | | | NoLan-Base | 86.23 | 92.76 | 78.60 | 85.10 |
| | | | NoLan-Plus | **87.40** | 93.15 | 80.73 | **86.50** |
| | | InstructBLIP-7B | Regular | 78.22 | 77.87 | 78.85 | 78.36 |
| | | | NoLan-Base | 83.97 | 90.73 | 75.67 | 82.52 |
| | | | NoLan-Plus | 84.00 | 92.29 | 74.20 | 82.26 |
| | | InstructBLIP-13B | Regular | 79.07 | 81.11 | 75.79 | 78.35 |
| | | | NoLan-Base | 85.57 | 87.78 | 81.40 | 85.04 |
| | | | NoLan-Plus | **85.90** | 88.77 | 82.20 | **85.36** |
| | Adversarial | LLaVA1.5-7B | Regular | 78.96 | 83.06 | 72.75 | 77.57 |
| | | | NoLan-Base | 83.00 | 88.73 | 75.60 | 81.64 |
| | | | NoLan-Plus | 83.60 | 89.31 | 76.33 | 82.31 |
| | | LLaVA1.5-13B | Regular | 80.00 | 84.46 | 73.53 | 78.62 |
| | | | NoLan-Base | 83.87 | 87.80 | 78.67 | 82.98 |
| | | | NoLan-Plus | **84.90** | 88.07 | 80.73 | **84.24** |
| | | InstructBLIP-7B | Regular | 75.84 | 74.30 | 79.03 | 76.59 |
| | | | NoLan-Base | 81.97 | 86.58 | 75.67 | 80.75 |
| | | | NoLan-Plus | 82.37 | 88.62 | 74.27 | 80.81 |
| | | InstructBLIP-13B | Regular | 76.57 | 77.00 | 75.79 | 76.39 |
| | | | NoLan-Base | 82.60 | 82.71 | 81.33 | 82.48 |
| | | | NoLan-Plus | **82.97** | 83.57 | 82.07 | **82.81** |

in the majority of evaluated scenarios, showcasing its enhanced effectiveness. While *sigmoid* achieves similar improvements on LLaVA, *tanh* demonstrates superior performance on Qwen-VL and InstructBLIP, underscoring its greater adaptability and broader generalization capability. This may be due to the faster convergence of the *tanh* function, allowing the moderation term to approach its upper bound more quickly under the imposed constraints, thereby more effectively mitigating the influence of linguistic priors.

## A.5 BENCHMARKING NOLAN AGAINST THE ICD BASELINE

As a member of the contrastive decoding family of methods, Instruction Contrastive Decoding (ICD) (Wang et al., 2024c) introduces a special mechanism into multimodal inference by injecting

Table 13: **Results of NoLan-Plus's variants on POPE (Li et al., 2023d).** *Sigmoid* refers to the use of the Sigmoid function as the processing term, while *Tanh* denotes the use of the Tanh function for the same purpose.The best performances within each setting are bolded.

| Dataset | Model | Function | Random | | Popular | | Adversarial | | All↑ |
|---|---|---|---|---|---|---|---|---|---|
| | | | Accuracy↑ | F1 Score↑ | Accuracy↑ | F1 Score↑ | Accuracy↑ | F1 Score↑ | |
| GQA | LLaVA1.5 | Sigmoid | 88.53 | 87.83 | 84.53 | 84.13 | 81.47 | 81.54 | 84.67 |
| | | Tanh | **88.57** | **87.88** | **84.57** | **84.31** | **81.50** | **81.88** | **84.79** |
| | Qwen-VL | Sigmoid | 86.83 | 86.44 | 82.20 | 82.42 | 80.10 | 80.76 | 83.13 |
| | | Tanh | **87.27** | **86.99** | **83.20** | **83.62** | **80.20** | **81.05** | **83.72** |
| | InstructBLIP | Sigmoid | 84.47 | 84.62 | 77.90 | 79.45 | 74.60 | 77.03 | 79.68 |
| | | Tanh | **86.13** | **85.23** | **81.13** | **80.92** | **78.10** | **78.41** | **81.65** |
| A-OKVQA | LLaVA1.5 | Sigmoid | 88.03 | 87.38 | 86.03 | 85.58 | 79.87 | 80.39 | 84.55 |
| | | Tanh | 88.00 | 87.30 | 85.70 | 85.22 | 79.47 | 80.01 | 84.28 |
| | Qwen-VL | Sigmoid | 88.63 | 88.04 | 87.87 | 87.41 | 80.93 | 81.44 | 85.72 |
| | | Tanh | **89.37** | **89.03** | **87.97** | **87.72** | **81.20** | **82.06** | **86.23** |
| | InstructBLIP | Sigmoid | 86.53 | 86.60 | 81.23 | 82.27 | 74.27 | 77.11 | 81.34 |
| | | Tanh | **88.20** | **87.55** | **84.57** | **84.32** | **78.43** | **79.24** | **83.72** |
| MSCOCO | LLaVA1.5 | Sigmoid | **87.03** | **85.46** | **85.63** | **84.14** | **83.63** | **82.36** | **84.71** |
| | | Tanh | 87.00 | 85.42 | 85.60 | 84.09 | 83.60 | 82.31 | 84.67 |
| | Qwen-VL | Sigmoid | 86.97 | 85.51 | 86.73 | 85.40 | 84.37 | 83.20 | 85.36 |
| | | Tanh | **88.10** | **87.00** | **87.43** | **86.40** | **84.90** | **84.07** | **86.32** |
| | InstructBLIP | Sigmoid | 85.53 | **84.54** | 83.33 | **82.60** | 80.63 | 80.34 | 82.83 |
| | | Tanh | **85.67** | 83.81 | **84.00** | 82.26 | **82.37** | **80.81** | **83.15** |

Table 14: Results of InstructBLIP on POPE (Li et al., 2023d). *Regular* decoding denotes direct sampling, *VCD* (Leng et al., 2024) indicates sampling from visual contrastive distribution, *ICD* (Wang et al., 2024c) expresses using Instruction Contrastive Decoding, while methods prefixed with *NoLan* refers to sampling from our proposed contrastive distribution $p_{nolan}$. The best performances within each setting are bolded.

| Dataset | Decoding | Random | | Popular | | Adversarial | |
|---|---|---|---|---|---|---|---|
| | | Accuracy ↑ | F1 Score ↑ | Accuracy↑ | F1 Score ↑ | Accuracy ↑ | F1 Score ↑ |
| GQA | Regular | 79.65 | 80.56 | 73.87 | 76.42 | 70.56 | 74.12 |
| | VCD | 83.69 | 84.16 | 78.57 | 80.17 | 75.08 | 77.53 |
| | ICD | 85.10 | 85.29 | 78.50 | 80.87 | 75.17 | 77.65 |
| | NoLan-Base (Ours) | 85.63 | 85.04 | 79.60 | 80.01 | 76.97 | 77.99 |
| | NoLan-Plus (Ours) | **86.13** | **85.23** | **81.13** | **80.92** | **78.10** | **78.41** |
| A-OKVQA | Regular | 80.91 | 81.86 | 76.19 | 78.17 | 70.71 | 75.56 |
| | VCD | 84.11 | 84.56 | 79.78 | 81.15 | 74.33 | 77.19 |
| | ICD | 85.82 | 86.29 | 81.64 | 83.32 | 74.42 | 78.48 |
| | NoLan-Base (Ours) | 87.87 | 87.46 | 83.60 | 83.76 | 77.33 | 78.79 |
| | NoLan-Plus (Ours) | **88.20** | **87.55** | **84.57** | **84.32** | **78.43** | **79.24** |
| MSCOCO | Regular | 80.71 | 80.41 | 78.22 | 78.36 | 75.84 | 76.59 |
| | VCD | 84.53 | 83.68 | 81.47 | 81.07 | 79.56 | 79.52 |
| | ICD | **86.43** | **85.61** | 82.93 | **82.55** | 80.87 | **80.84** |
| | NoLan-Base (Ours) | 86.07 | 84.45 | 83.97 | 82.52 | 81.97 | 80.75 |
| | NoLan-Plus (Ours) | 85.67 | 83.81 | **84.00** | 82.26 | **82.37** | 80.81 |

carefully crafted disturbance instructions during decoding. According to its process, ICD augments the input with a misleading prompt (e.g., "You are a confused object detector") to intentionally increase alignment uncertainty. This yields two distributions: one conditioned on the standard instruction and another on the disturbed version. By subtracting the latter from the former, ICD aims to suppress hallucinated concepts that are overactivated by visual priors, thus enhancing prediction robustness.

In this section, we provide a supplementary evaluation of the ICD baseline and compare its performance with our proposed method built upon InstructBLIP. Table 14 presents a detailed comparison across three datasets (GQA, A-OKVQA, and MSCOCO), covering random, popular, and adversarial question categories. The results demonstrate that while ICD shows clear improvements over standard decoding and VCD (Leng et al., 2024), our NoLan variants consistently outperform it across most settings.

### A.6 SUPPLEMENTARY EXPERIMENTS

**MM-Vet.** In addition to using POPE (Li et al., 2023d) for evaluation, we incorporate open-ended questions assessed with an LLM-based evaluator to deliver a more thorough and comprehensive analysis of its performance. MM-Vet is an advanced benchmark designed to evaluate the capabilities of Large Multimodal Models (LMMs) in tackling complex multimodal tasks (Yu et al., 2023). It defines

Table 15: **MM-Vet (Yu et al., 2023) evaluation results regarding each core VL capability.** All the numbers are presented in % and the full score is 100%. Our NoLan can improve performance for different models.

| Model | Rec | OCR | Know | Gen | Spat | Math | Total |
|---|---|---|---|---|---|---|---|
| LLaVA1.5-7B (Liu et al., 2024a) | – | – | – | – | – | – | 31.1 |
| LLaVA1.5-7B NoLan-Base | 36.2 | 26.5 | 21.3 | 23.3 | 33.5 | 7.7 | 33.0±0.1 |
| **LLaVA1.5-7B NoLan-Plus** | 38.0 | 25.7 | 18.8 | 24.2 | 31.8 | 7.7 | **33.3±0.2** |
| LLaVA1.5-13B (Liu et al., 2024a) | – | – | – | – | – | – | 36.1 |
| LLaVA1.5-13B NoLan-Base | 42.2 | 29.8 | 27.3 | 28.2 | 35.2 | 14.2 | 37.6±0.2 |
| **LLaVA1.5-13B NoLan-Plus** | 41.8 | 31.4 | 24.9 | 26.0 | 36.6 | 15.4 | **38.3±0.2** |
| InstructBLIP-7B | 30.7 | 16.2 | 15.3 | 13.2 | 22.3 | 7.7 | 25.2±0.0 |
| InstructBLIP-7B NoLan-Base | 32.8 | 13.0 | 14.0 | 14.3 | 17.1 | 4.2 | 25.7±0.1 |
| **InstructBLIP-7B NoLan-Plus** | 35.1 | 13.6 | 17.8 | 18.7 | 16.7 | 3.8 | **27.0±0.1** |
| InstructBLIP-13B | 25.1 | 12.8 | 10.5 | 8.5 | 18.6 | 5.8 | 21.2±0.3 |
| InstructBLIP-13B NoLan-Base | 31.7 | 12.5 | 15.7 | 9.5 | 19.9 | 3.5 | 25.4±0.2 |
| **InstructBLIP-13B NoLan-Plus** | 30.9 | 18.6 | 12.7 | 8.7 | 22.4 | 11.5 | **26.7±0.1** |
| Qwen-VL | 33.7 | 27.7 | 18.5 | 10.1 | 33.2 | 11.2 | 33.7±0.1 |
| Qwen-VL NoLan-Base | 36.0 | 26.9 | 17.5 | 9.0 | 33.0 | 7.7 | 34.5±0.1 |
| **Qwen-VL NoLan-Plus** | 36.8 | 26.5 | 21.8 | 13.6 | 32.6 | 7.7 | **35.2±0.2** |

Table 16: **MM-Vet (Yu et al., 2023) evaluation results regarding each capability integration.** Our NoLan can improve model performance for different models.

| Model | Rec Know Gen | Rec | OCR Spat | OCR Spat Math | Rec Spat | OCR | OCR Math | Rec Know | Rec OCR Know Gen | Rec OCR Gen Spat | Rec OCR Spat | Rec OCR | OCR Know Spat | Rec Know Spat | OCR Gen Spat | Rec OCR Spat Math | Total |
|---|---|---|---|---|---|---|---|---|---|---|---|---|---|---|---|---|---|
| LLaVA1.5-7B (Liu et al., 2024a) | – | – | – | – | – | – | – | – | – | – | – | – | – | – | – | – | 31.1 |
| LLaVA1.5-7B NoLan-Base | 21.1 | 62.2 | 34.6 | 14.3 | 65.8 | 40.8 | 0.0 | 27.8 | 16.8 | 43.8 | 14.3 | 50.0 | 33.3 | 0.0 | 37.0 | 0.0 | 33.0±0.1 |
| **LLaVA1.5-7B NoLan-Plus** | 22.3 | 68.9 | 29.8 | 14.3 | 66.7 | 40.8 | 0.0 | 5.6 | 52.2 | 12.5 | 14.3 | 75.0 | 16.7 | 0.0 | 20.0 | 0.0 | 33.3±0.2 |
| LLaVA1.5-13B (Liu et al., 2024a) | – | – | – | – | – | – | – | – | – | – | – | – | – | – | – | – | 36.1 |
| LLaVA1.5-13B NoLan-Base | 27.5 | 70.0 | 28.5 | 26.4 | 58.3 | 47.3 | 0.0 | 26.7 | 12.5 | 54.2 | 17.1 | 87.5 | 50.0 | 50.0 | 50.0 | 0.0 | 37.6±0.2 |
| **LLaVA1.5-13B NoLan-Plus** | 23.5 | 73.5 | 34.7 | 28.6 | 58.3 | 53.3 | 0.0 | 38.9 | 51.2 | 16.2 | 14.3 | 75.0 | 16.7 | 50.0 | 40.0 | 0.0 | 38.3±0.2 |
| InstructBLIP-7B | 12.0 | 63.7 | 13.4 | 14.3 | 41.7 | 15.8 | 0.0 | 27.8 | 33.0 | 5.2 | 14.3 | 62.5 | 50.0 | 50.0 | 5.0 | 0.0 | 25.2±0.0 |
| InstructBLIP-7B NoLan-Base | 14.3 | 73.7 | 13.5 | 7.9 | 45.8 | 19.0 | 0.0 | 22.2 | 21.5 | 11.5 | 14.3 | 50.0 | 0.0 | 0.0 | 0.0 | 0.0 | 25.7±0.1 |
| **InstructBLIP-7B NoLan-Plus** | 17.9 | 73.5 | 11.5 | 7.1 | 41.7 | 18.5 | 0.0 | 27.8 | 31.2 | 16.8 | 14.3 | 50.0 | 0.0 | 0.0 | 0.0 | 0.0 | **27.0±0.1** |
| InstructBLIP-13B | 8.0 | 58.1 | 19.2 | 10.7 | 33.3 | 16.7 | 0.0 | 27.8 | 17.5 | 4.2 | 14.3 | 25.0 | 0.0 | 50.0 | 5.0 | 0.0 | 21.2±0.3 |
| InstructBLIP-13B NoLan-Base | 10.4 | 67.6 | 13.5 | 0.0 | 58.3 | 19.8 | 8.2 | 44.4 | 5.0 | 9.8 | 28.6 | 25.0 | 33.3 | 50.0 | 0.0 | 0.0 | 25.4±0.2 |
| **InstructBLIP-13B NoLan-Plus** | 6.4 | 72.7 | 15.4 | 14.3 | 50.0 | 23.3 | 9.1 | 38.9 | 21.5 | 15.2 | 14.3 | 50.0 | 66.7 | 0.0 | 4.0 | 0.0 | **26.7±0.1** |
| Qwen-VL | 11.5 | 76.9 | 42.3 | 14.3 | 54.2 | 47.5 | 8.2 | 42.2 | 7.5 | 1.2 | 14.3 | 50.0 | 100.0 | 50.0 | 14.0 | 0.0 | 33.7±0.1 |
| Qwen-VL NoLan-Base | 9.5 | 83.2 | 44.2 | 14.3 | 59.8 | 47.5 | 0.0 | 50.0 | 16.5 | 0.5 | 14.3 | 57.5 | 66.7 | 50.0 | 0.0 | 0.0 | 34.5±0.1 |
| **Qwen-VL NoLan-Plus** | 15.0 | 80.5 | 40.4 | 7.1 | 62.5 | 50.0 | 9.1 | 42.7 | 2.8 | 14.8 | 14.3 | 32.5 | 100.0 | 50.0 | 10.0 | 0.0 | **35.2±0.2** |

16 novel tasks of significant importance, derived from six core visual-language (VL) capabilities, and employs an LLM-based evaluator to assess the open-ended outputs of LMMs. To demonstrate the effectiveness of NoLan in open-ended generation tasks, we conducted a comprehensive evaluation using the MM-Vet benchmark and its GPT-4 aided evaluator. This benchmark can test NoLan's performance in scenarios requiring nuanced and contextually accurate multimodal understanding.

As shown in Table 15, NoLan consistently outperforms regular decoding across both 7B and 13B models, highlighting its ability to enhance the open-ended generation capabilities of LVLMs. Notably, the findings also suggest that NoLan's effectiveness scales with larger model sizes, delivering sustained improvements as models increase in complexity. For example, NoLan-Plus improves the performance of LLaVA-1.5 7B from 31.1 to 33.3, while the 13B model increases from 36.1 to 38.3. Additionally, as shown in Table 16, most capability integrations exhibit growth. For instance, the combination of "Rec" and "Spat" shows an increase of up to 8.3%. Furthermore, the results demonstrate that mitigating object hallucinations can positively impact open-ended generation capabilities.

This result is not entirely unexpected, as the original model often generates content with hallucinatory effects in open-ended tasks. Previous experiments have demonstrated NoLan's effectiveness in reducing hallucinations, reinforcing its ability to address this issue. Importantly, unlike the binary classification setting in POPE, the diversity of the generated content plays a crucial role in this evaluation. Despite this added complexity, NoLan consistently achieves higher evaluation scores, demonstrating its ability to mitigate hallucinations while preserving the diversity of the model's output. This balance allows the model to excel in open-ended question responses, showcasing NoLan's capability to enhance both accuracy and content richness.

**MMHAL-BENCH.** MMHAL-BENCH (Sun et al., 2023) is a 96-pair benchmark that tests hallucination in large multimodal models across eight error types: wrong object attributes, nonexistent objects,

Table 17: Results on MMHalBench (Sun et al., 2023) for different decoding strategies. NoLan variants improve the overall score and decrease the hallucination rate.

| Decoding | MMHalBench | | | | | | | | |
| | Overall Score↑ | Hallucination Rate↓ | Attribute | Adversarial | Comparison | Counting | Relation | Environment | Holistic | Other |
|---|---|---|---|---|---|---|---|---|---|---|
| Regular | 1.55 | 76% | 1.33 | 0 | 1.83 | 1.17 | 2.00 | 2.58 | 1.67 | 1.83 |
| NoLan-Base | 1.85 | 75% | 3.42 | 1.58 | 1.42 | 1.42 | 1.75 | 3.33 | 0.83 | 1.08 |
| NoLan-Plus | 2.29 | 68% | 3.42 | 3.25 | 1.33 | 1.58 | 1.58 | 3.83 | 1.33 | 2.00 |

faulty comparisons, counting errors, spatial mistakes, false environment inferences, misleading holistic descriptions, and misrecognition of text or icons.

As shown in Table 17, we evaluate different decoding strategies on LLaVA-1.5 (Liu et al., 2024a) using this challenging benchmark. The metrics include the overall score (higher is better) and hallucination rate (lower is better), as well as category-wise breakdowns. Both NoLan variants outperform regular decoding: NoLan-Base improves the overall score from 1.55 to 1.85, while NoLan-Plus further increases it to 2.29 and reduces the hallucination rate from 76% to 68%.

These results highlight that suppressing language priors not only boosts semantic alignment but also reduces vulnerability to hallucination across diverse categories, with NoLan-Plus showing the strongest robustness against visual misinterpretation.

Table 18: Results on HallusionBench (Guan et al., 2024) for different decoding strategies. NoLan variants improve overall accuracy (aAcc) and category-specific metrics.

| Decoding | HallusionBench | | | | |
| | qAcc | fAcc | easyaAcc | hardaAcc | aAcc |
|---|---|---|---|---|---|
| Regular | 14.2857 | 15.6069 | 37.1429 | 38.8372 | 43.4898 |
| NoLan-Base | 15.1648 | 17.9191 | 45.0549 | 35.5814 | 46.5899 |
| NoLan-Plus | 18.6813 | 19.6532 | 43.9560 | 40.6977 | 47.4756 |

**HallusionBench.** HallusionBench (Guan et al., 2024) is a recently proposed diagnostic benchmark specifically designed to probe and quantify the failure modes of large vision-language models (LVLMs) in image-context reasoning. It consists of 1129 handcrafted visual-question-answer (VQA) pairs, built upon 346 distinct visual figures—including original and human-edited images—covering a wide range of domains such as geometry, food, statistics, maps, and visual illusions. Each question pair is designed to reveal inconsistencies or hallucinations in model predictions, going beyond traditional accuracy metrics to expose deeper reasoning flaws.

As shown in Table 18, we evaluate different decoding strategies on LLaVA-1.5 (Liu et al., 2024a) using this challenging benchmark. The metrics include qAcc (Question Pair Accuracy), fAcc (Figure Accuracy) over both easy and hard examples. Both NoLan variants outperform regular decoding across all metrics. In particular, NoLan-Plus achieves the highest question accuracy (qAcc: 18.68) and overall accuracy (aAcc: 47.48), suggesting improved robustness against hallucinations and visual misinterpretation.

These results highlight that suppressing language priors not only enhances semantic alignment but also reduces model vulnerability to visually deceptive or noisy contexts, especially on hard cases (hardaAcc: 40.70). HallusionBench thus provides critical insights into the nuanced failure modes of LVLMs and demonstrates the effectiveness of contrastive decoding in mitigating them.

Table 19: CircularEval results on MMBench (Liu et al., 2024d) test set (L-2 abilities). NoLan variants improve overall and category-specific metrics.

| Decoding | MMBench | | | | | | |
| | Overall | AR | CP | FP-C | FP-S | LR | RR |
|---|---|---|---|---|---|---|---|
| Regular | 63.4 | 77.6 | 70.0 | 57.7 | 68.0 | 33.2 | 56.2 |
| NoLan-Base | 64.6 | 76.0 | 77.1 | 56.7 | 66.3 | 33.0 | 53.6 |
| NoLan-Plus | 65.8 | 74.7 | 77.5 | 55.1 | 67.1 | 38.7 | 60.2 |

**MMBench.** MMBench (Liu et al., 2024d) is a systematically constructed benchmark designed to evaluate a wide range of vision-language capabilities across 20 distinct ability dimensions, such as

object localization, commonsense reasoning, and social understanding. Each ability is uniformly represented by over 125 multiple-choice questions, enabling balanced and fine-grained assessment. To address inconsistencies caused by VLMs' limited instruction-following capabilities, the benchmark employs GPT-4 as a robust choice extractor, achieving 91.5% alignment with human judgment.

To further improve evaluation robustness, MMBench introduces CircularEval—a strategy designed to reduce bias and variance in performance assessment by aggregating multiple sampling and evaluation rounds. This method emphasizes consistency across ability dimensions and mitigates artifacts from instruction misalignment or label mismatch.

As shown in Table 19, we evaluate decoding strategies on LLaVA-1.5 (Liu et al., 2024a) using the CircularEval protocol. Both NoLan variants outperform regular decoding in overall accuracy and several reasoning-specific dimensions. In particular, NoLan-Plus achieves the highest overall score (65.8) and shows notable improvements in Coarse Perception (CP: 77.5) and Relation Reasoning (RR: 60.2), alongside gains in Logical Reasoning (LR: 38.7). These dimensions—abbreviated in Table 19 as CP, RR, and LR—correspond to L-2 level cognitive skills, which demand deeper visual-semantic understanding.

These results indicate that suppressing language priors not only benefits general performance but also enhances high-level reasoning under rigorous evaluation settings like CircularEval. Moreover, the improved consistency across fine-grained and relational tasks suggests better grounding and reduced over-reliance on textual shortcuts.

Table 20: Results on MathVision (Wang et al., 2024a) for different decoding strategies. NoLan variants improve overall and most sub-categories, such as Algebra (Alg), Geometry (e.g., Angle, Area), and Logical reasoning (Log).

| Decoding | MathVision | | | | | | | | | | | | | | | | |
|---|---|---|---|---|---|---|---|---|---|---|---|---|---|---|---|---|---|
| | ALL | Alg | AnaG | Ari | CombG | Comb | Cnt | DescG | GrphT | Log | Angle | Area | Len | SolG | Stat | Topo | TransG |
| Random Chance | 7.17 | 1.50 | 11.90 | 7.10 | 9.70 | 4.80 | 6.00 | 22.10 | 1.10 | 7.60 | 0.60 | 9.40 | 6.70 | 8.20 | 8.60 | 13.00 | 7.10 |
| Regular | 8.52 | 7.00 | 7.10 | 10.70 | 7.10 | 4.80 | 10.50 | 7.70 | 10.00 | 9.20 | 15.60 | 10.20 | 9.80 | 5.30 | 8.60 | 4.40 | 4.80 |
| NoLan-Base | 9.34 | 5.22 | 4.76 | 5.71 | 11.36 | 7.14 | 8.96 | 13.46 | 14.44 | 7.56 | 13.29 | 10.60 | 10.02 | 6.15 | 17.24 | 4.35 | 10.71 |
| NoLan-Plus | 9.84 | 6.96 | 8.33 | 7.14 | 11.04 | 6.55 | 5.97 | 17.31 | 17.78 | 7.56 | 13.29 | 8.60 | 9.80 | 9.43 | 13.79 | 13.04 | 13.10 |

**MathVision.** MathVision (Wang et al., 2024a) (MATH-V) is a curated benchmark designed to assess the mathematical reasoning capabilities of large multimodal models in visually grounded settings. The dataset consists of 3,040 high-quality visual math problems spanning 16 mathematical disciplines and 5 difficulty levels, covering topics such as algebra, combinatorial geometry, topology, and logic. Problems are sourced from 19 official math competitions and are annotated and verified by domain experts to ensure uniqueness and correctness of answers. The benchmark contains both multiple-choice and open-ended formats, requiring models to perform fine-grained multimodal understanding and symbolic reasoning.

As illustrated in Table 20, we compare decoding strategies on LLaVA-1.5 (Liu et al., 2024a) across all subject areas. Both NoLan variants significantly outperform the regular baseline in overall performance (ALL), with NoLan-Plus achieving the best accuracy (9.84%). Improvements are especially prominent in core areas such as Algebra (Alg: 6.96), Graph Theory (GrphT: 17.78), and metric geometry - angle (Angle: 13.29), all of which require both precise visual perception and subject-specific mathematical reasoning.

These results demonstrate that suppressing language priors helps reduce superficial biases and encourages more deliberate reasoning. MathVision thus reveals the benefits of contrastive decoding in tackling symbolically grounded, visually rich tasks where hallucinations and template-like answers are common failure modes for conventional VLMs.

## A.7 CONTRASTING NoLan WITH ATTENTION-BASED APPROACHES

While our main analysis focuses on contrastive decoding strategies, several recent methods adopt alternative training-free techniques to mitigate hallucinations by intervening in the attention mechanism. Among them, Pay Attention to Image (PAI)(Liu et al., 2024c) and OPERA(Huang et al., 2024) stand out as representative and competitive approaches.

Table 21: Comparison between NoLan and attention-based methods

| Decoding | MSCOCO of POPE-*random* | |
| --- | --- | --- |
| | **Accuracy** | **F1 Score** |
| Regular | 83.29 | 81.33 |
| OPERA (Huang et al., 2024) | – | 85.40 |
| PAI (Liu et al., 2024c) | 86.33 | 85.89 |
| NoLan-Base (Ours) | 87.80 | 85.60 |
| **NoLan-Plus (Ours)** | **88.80** | **86.70** |

PAI operates by amplifying attention weights directed toward image tokens during inference. It adjusts the self-attention heads in the decoder layers to emphasize image regions in their original direction, thereby reducing reliance on language priors. In addition, PAI constructs auxiliary textual prompts (comprising instructions and historical responses) and subtracts their logits from the image-conditioned logits. This dual intervention strategy encourages more image-grounded reasoning while suppressing text inertia. Importantly, PAI is fully training-free and directly targets two key issues: image neglect and language dominance.

OPERA, on the other hand, addresses the over-trust phenomenon in beam search decoding. It introduces a column-wise metric over the attention map to detect knowledge aggregation patterns that correlate with hallucination. A penalty score is integrated with the logits during candidate selection, disfavoring over-trusted tokens. Additionally, OPERA employs a retrospection-reallocation mechanism that can roll back to previous decoding positions if over-trust is detected, enabling the model to reallocate attention and choose alternative candidates.

As shown in Table 21, we compare these methods on LLaVA-1.5 (Liu et al., 2024a) using the MSCOCO of the POPE benchmark. Both PAI and OPERA achieve strong results, with F1 scores of 85.89 and 85.40, respectively. Our NoLan-Plus further improves on these with the highest accuracy (88.80) and F1 score (86.70), demonstrating that contrastive decoding with language prior suppression remains a highly effective strategy.

These results suggest that while attention-based methods offer promising avenues, contrastive decoding offers a more general and robust framework for hallucination mitigation, especially when the distributional shift is carefully controlled by leveraging the difference between dual forward outputs.

### A.8 QWEN-VL SERIES

Table 22: Results of Qwen2-VL (Wang et al., 2024b) and Qwen2.5-VL (Bai et al., 2025) on POPE (Li et al., 2023d).

| Decoding | MSCOCO of POPE-*random* | | | |
| --- | --- | --- | --- | --- |
| | **Accuracy** | **Precision** | **Recall** | **F1 Score** |
| | *Qwen2-VL-2B* | | | |
| Regular | 71.27 | 76.76 | 61.00 | 67.98 |
| NoLan-Base | 73.93 | 85.83 | 57.33 | 68.75 |
| NoLan-Plus | **77.67** | 79.64 | 74.33 | **76.90** |
| | *Qwen2-VL-7B* | | | |
| Regular | 87.27 | 96.58 | 77.27 | 85.85 |
| NoLan-Base | 88.90 | 98.18 | 79.27 | 87.72 |
| NoLan-Plus | **89.80** | 97.38 | 81.80 | **88.91** |
| | *Qwen2.5-VL-3B* | | | |
| Regular | 87.27 | 91.78 | 81.87 | 86.54 |
| NoLan-Base | 88.57 | 97.85 | 78.87 | 87.34 |
| NoLan-Plus | **90.67** | 93.70 | 87.20 | **90.33** |
| | *Qwen2.5-VL-7B* | | | |
| Regular | 83.70 | 99.22 | 67.93 | 80.65 |
| NoLan-Base | 87.40 | 98.36 | 76.07 | 85.79 |
| NoLan-Plus | **88.63** | 92.89 | 83.67 | **88.04** |

Qwen2-VL (Wang et al., 2024b) and its successor Qwen2.5-VL (Bai et al., 2025) are recent multimodal large language model families that unify image, text, and video processing through a dynamic resolution mechanism and multimodal rotary position embedding (M-RoPE). The series scales across parameter sizes from 2B to 72B, with Qwen2.5 introducing architectural refinements for stronger visual–language alignment.

Table 22 reports POPE results with NoLan. On Qwen2-VL-2B, NoLan-Plus improves F1 from 67.98 to 76.90, while on Qwen2-VL-7B it raises F1 from 85.85 to 88.91. Similar trends hold for Qwen2.5-VL: NoLan-Plus boosts F1 from 86.54 to 90.33 on the 3B model and from 80.65 to 88.04 on the 7B model, with substantial recall gains. These consistent improvements across scales and generations demonstrate the robustness of NoLan in enhancing visual grounding.

These results highlight the generality and scalability of our contrastive decoding approach: even when integrated with advanced architectures like Qwen2-VL and Qwen2.5-VL, NoLan continues to effectively suppress language priors and enhance grounding, particularly in challenging settings like POPE where precise visual grounding is essential.

## A.9    CONSUMPTION OF INFERENCE

Table 23: Inference efficiency comparison of contrastive decoding strategies.

| Decoding | Seconds per Token ↓ | Memory Usage (GB, 50 tokens) ↓ |
|---|---|---|
| Regular | 0.4579 | 13.57 |
| VCD | 0.7537 | 15.09 |
| VDD | 0.7359 | 15.09 |
| NoLan-Base | 0.6075 | 13.59 |
| NoLan-Plus | 0.6277 | 13.59 |

We compare the inference efficiency of NoLan and contrastive decoding baselines (VCD, VDD) in terms of computation time and memory usage. As shown in Table 23, VCD and VDD require two forward passes over inputs $(v, x)$ and $(v', x)$, along with additional post-processing using adaptive plausibility constraints. In contrast, NoLan simplifies this process by using only $(v, x)$ and $(x)$ as inputs. NoLan-Base requires no post-processing, and NoLan-Plus adds only a lightweight KL divergence computation, making both significantly more efficient.

Empirical results on LLaVA-v1.5-7B with a Titan RTX 24GB GPU confirm the efficiency of NoLan: among contrastive decoding methods, NoLan-Base achieves the fastest inference speed (0.6075 seconds per token) and the lowest memory usage (13.59 GB for 50 tokens). In comparison, VCD and VDD are both slower and more memory-intensive (15.09 GB), underscoring NoLan's practical advantages in latency and resource efficiency.

## A.10    ETHICS AND REPRODUCIBILITY STATEMENTS

**Ethics statement.** Our research adheres to the ICLR Code of Ethics. The primary focus of our work is to mitigate object hallucinations in Large Vision-Language Models (LVLMs). Object hallucination, a phenomenon where models generate text describing objects that are either mismatched or entirely absent in an image, poses a significant ethical concern. Such fabrications can lead to the spread of misinformation and reduce the reliability of AI systems in critical applications. Our proposed method, NoLan, contributes to the development of more trustworthy and factual AI by directly addressing this issue. By suppressing the language priors that we identify as a principal cause of hallucinations, NoLan improves the accuracy and faithfulness of LVLM outputs. This can have positive societal benefits by making these models safer and more reliable for public use. The datasets and models used in our experiments are publicly available, and our research does not involve any personally identifiable information or sensitive data. We will make our code publicly available to encourage transparency and allow for further research in this area. We are not aware of any direct negative social impacts or ethical concerns arising from our work. We believe that by improving the factuality of LVLMs, our work represents a positive step towards more ethical and responsible AI.

**Reproducibility statement.** To ensure full reproducibility, we will make our complete source code publicly available. This repository contains the implementation of our NoLan framework, alongside

all scripts necessary to replicate our experiments and evaluations against the reported baselines (Regular, VCD, M3ID, and VDD). Our experiments are conducted on publicly accessible LVLMs, including the LLaVA-1.5, InstructBLIP, and Qwen-VL series, using widely-adopted benchmarks. Specifically, we use POPE, MME, and LLaVA-Bench in the main paper, with extended evaluations on MM-Vet, MMHAL-BENCH, and MMBench, among others, in the appendix. As detailed in Section 4, our experimental setup, including dataset-specific configurations, aligns with prior work for fair comparison. For our mechanism in NoLan-Plus variant, a complete theoretical proof is also provided in the appendix. This comprehensive release is intended to allow the community to easily verify our findings and build upon our work.

### A.11 MORE CASE STUDIES

To further validate the impact and effectiveness of our proposed NoLan-Plus on open-ended generation tasks, we conduct additional case studies on the LLaVA-bench. Figure 5 provides further instances of hallucination corrections by NoLan-Plus. In the examples presented, objects such as "*thinking face*" and "*question mark*," which are commonly linked to the ground truth object "*funny joke*," mistakenly appear as hallucinations in the generated output. In contrast, applying NoLan-Plus substantially reduces these hallucinations while maintaining the consistency and richness of the generated text. This highlights NoLan-Plus's ability to produce outputs that are more faithfully aligned with the visual input, without compromising informativeness. Additional examples are provided in Figure 6.

1404
1405
1406
1407
1408
1409
1410
1411
1412
1413
1414
1415
1416
1417
1418
1419
1420
1421
1422
1423
1424
1425
1426
1427
1428
1429
1430
1431
1432
1433
1434
1435
1436
1437
1438
1439
1440
1441
1442
1443
1444
1445
1446
1447
1448
1449
1450
1451
1452
1453
1454
1455
1456
1457

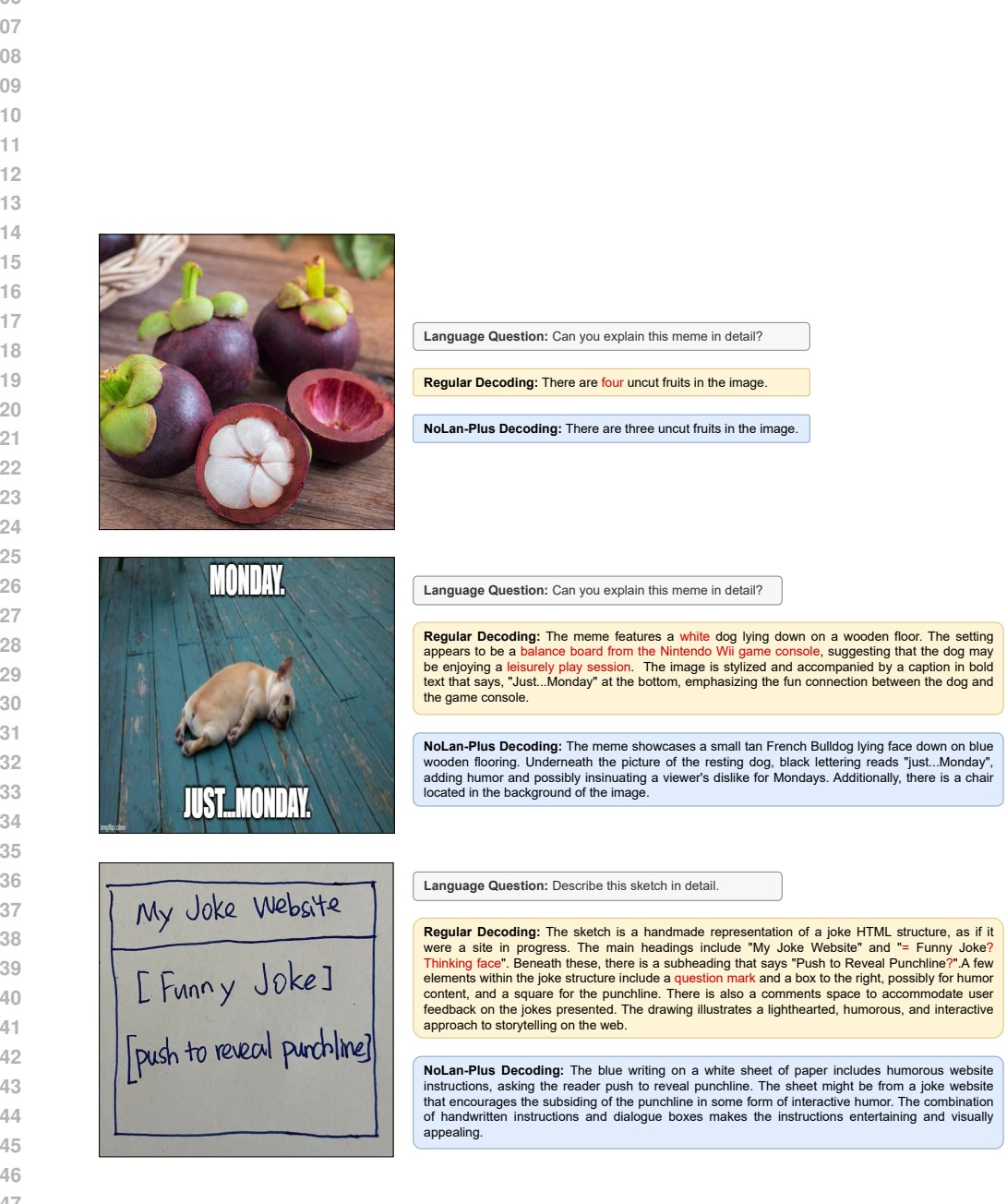

Figure 5: More examples from LLaVA-Bench of our proposed NoLan-Plus for hallucination corrections. Hallucinated objects from LVLM's regular decoding are highlighted in red.

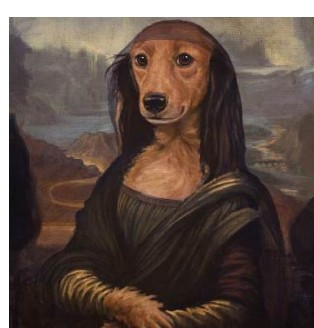

**Language Question:** Can you explain this meme in detail?

**Regular Decoding:** The image features a dog dressed up in a flowing costume, resembling a painting of a young woman. The furry canine, which appears to be a large brown and white dog, is sitting on a chair in a large living room. It seems to be posing for the unique human-like portrait. The room features additional furnishings such as a second small chair, located at the right side of the image, and a rock formation in the background on the right side. Overall, the scene is quite unusual with the focus on the dressed-up dog.

**NoLan-Plus Decoding:** The painting features a canine portrait dressed as Mona Lisa, looking to its side with one paw crossed. It seems like the artist painted the pit bull masquerading as a religious figure. The large pit bull/female character is surrounded by mountains and a river, adding depth to the background. The scene has a whimsical touch, combining human imagery with a dog, creating a unique, blended composition.

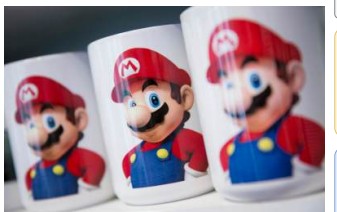

**Language Question:** Can you explain this meme in detail?

**Regular Decoding:** The image shows a collection of four coffee mugs placed horizontally on a shelf. Each mug is decorated differently, but they all share a common theme - they are designed with cartoon characters resembling Super Mario. The characters are positioned around the perimeter of each mug, giving them a protruding appearance. There is a clear separation between the four individual mugs, with each one having its own distinct arrangement of Super Mario characters displayed.

**NoLan-Plus Decoding:** The image features three coffee mugs each portraying Mario, the popular Nintendo character. The mugs closely stacked next to each other depict the same animated face design, making the scene visually indistinguishable. The collective presence of the close-up mugs covers objects such as another coffee mug and some ears, adding to the feeling of uniformity throughout the scene.

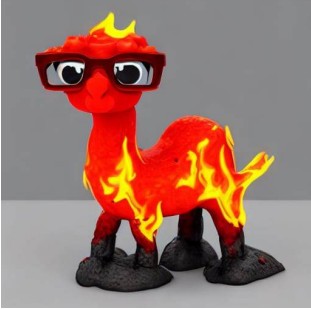

**Language Question:** Describe this sketch in detail.

**Regular Decoding:** The image features a unique figurine of an anthropomorphic red and yellow flamingo, which appears to be on fire, in a large glass case. The flamingo, wearing red and green glasses for fun, is placed on a shelf in the display. The glass case surrounds the flamingo, emphasizing and protecting it from potential damage. This whimsical display adds a touch of fun to the room where it is found.

**NoLan-Plus Decoding:** The image features a whimsical figurine of a flaming, toy animal with glasses made out of plastic. The unique creation portrays an odd and eye-catching design. The horse has a large eyeglasses and its legs are covered in flames, adding to its whimsical appearance. Adding a fun touch to the scene, the toy animal appears to be sitting on a pedestal, further enhancing its exhibition-like setting.

Figure 6: More examples from LLaVA-Bench of our proposed NoLan-Plus for hallucination corrections. Hallucinated objects from LVLM's regular decoding are highlighted in red.

