# OpenReview forum: "NoLan: Mitigating Object Hallucinations in Large Vision-Language Models via Dynamic Suppression of Language Priors"
_ICLR.cc/2026/Conference — ICLR 2026 Conference Withdrawn Submission_

### Official Review · Reviewer_AoHd · 2025-10-25

**Soundness:** 2
**Presentation:** 2
**Contribution:** 1
**Rating:** 2
**Confidence:** 5

**Summary:**

NoLan aims to mitigate object hallucination in VLMs. The core assumption is: if the next-token *distribution during original decoding with an image* is similar to the *distribution when decoding without the image*, the model is likely not grounded on the visual input. Operationally, the method measures the KL divergence between these two distributions and applies contrastive decoding more aggressively when the distributions are closer (low KL), and more weakly when they are farther apart (high KL).

**Strengths:**

1. The analysis comparing forward vs. reverse KL is meaningful. Since grounded visual information enters as an additional modality, validating grounding via forward KL (from the “no-image” to the “with-image” distribution) is a reasonable choice.

2. Modulating the strength of contrastive decoding as a function of the measured KL divergence is intuitive.

**Weaknesses:**

The paper appears highly vulnerable to the well-known drawbacks [1,2,3,4] of contrastive decoding, which can suppress the language prior, often harming text quality and logical coherence.

1. **(Very major)** The language prior embodies an LLM’s fluency and reasoning. Any method that suppresses it must quantify degradation in text quality (e.g., perplexity, LLM-as-judge, or human evaluation). Without such analysis, the approach will be hard to adopt broadly.

2. **(Very major)** When the KL divergence is small (i.e., the VLM and LLM distributions are already very close) but the method increases contrastive pressure, **the next-token distribution can easily collapse!** Under top-k or nucleus sampling, this is likely to produce linguistically broken outputs. The paper should at least analyze this failure mode and justify why KL-aware control does not over-regularize in such cases. Moreover, as shown in Appendix 3, the two distributions quickly converge as sequences grow longer; for tasks that require longer generations, the method may substantially degrade performance, limiting its applicability.

3. **(Major)** The claim that the “no-image” setting is the right reference distribution lacks sufficient analysis. Prior work (e.g., VCD, M3ID) offers differing views with noisy input or other styles. Given that VLMs are tuned to exploit vision-encoder inputs, it is not obvious that removing the image should approximate the LLM’s standalone distribution in a principled way. This needs a clearer theoretical or empirical justification.

4. **(major to moderate)** I strongly recommend including SumGD [4] or ClearSight [3] as baselines. Both methods reduce object hallucination while preserving text quality. Comparing against such approaches would clarify the trade-off your method makes between hallucination suppression and linguistic quality.

Overall, the paper’s two stated contributions—(1) moving toward the LLM distribution by omitting the image and (2) KL-aware control of contrastive decoding—are undermined by the three weaknesses above, which appear critical.

[1] GECOR: A Greedy-based Contrastive Decoding Strategy for Faithful and Coherent Text Generation

[2] Cross-Image Contrastive Decoding

[3] ClearSight: Visual Signal Enhancement for Object Hallucination Mitigation in Multimodal Large language Models

[4] Mitigating Hallucinations in Large Vision-Language Models via Summary-Guided Decoding

**Questions:**

Please see the weakness.

---

### Official Review · Reviewer_3Q4h · 2025-10-26

**Soundness:** 2
**Presentation:** 3
**Contribution:** 2
**Rating:** 4
**Confidence:** 4

**Summary:**

This paper presents NoLan (No-Language-Hallucination Decoding), a simple and training-free framework for mitigating object hallucinations in large vision-language models (LVLMs). Through analysis, the authors find that hallucinations mainly stem from language priors in the decoder rather than errors in the vision encoder. To address this, NoLan contrasts multimodal and text-only output distributions and dynamically suppresses language priors using a Kullback–Leibler divergence–based modulation. Two variants are proposed: NoLan-Base, with a fixed modulation rate, and NoLan-Plus, with an adaptive rate. Experiments on benchmarks such as POPE, MME, and LLaVA-Bench show that NoLan consistently reduces hallucinations and improves accuracy and F1 scores across LLaVA, Qwen-VL, and InstructBLIP, validating its generality and efficiency.

**Strengths:**

1. The motivation of this work is reasonable, as text-prior bias is indeed an important cause of hallucinations in large vision-language models. The proposed research to address this issue is therefore meaningful and valuable.
2. The writing is generally good and easy to follow.
3. NoLan-Plus is reasonable and a useful contribution.

**Weaknesses:**

1. The comparative experiments are not sufficiently comprehensive. For example, on the POPE benchmark, the authors only compared their method with VCD while overlooking many more recent approaches.
2. While language-prior bias is indeed a reasonable explanation for hallucinations in LVLMs, it may not be a novel finding of this paper. Many prior studies, such as [a, b], have already thoroughly investigated this issue.
3. The proposed method uses the output from text-only inputs as the contrastive reference, which confuses me. Without any visual information, the response generated from a text-only input seems meaningless. This is especially concerning for benchmarks like POPE, where the input is simply a question such as “Is there a [class] in the image?”. Without any visual information, the model’s response could be essentially random. In such a case, does performing contrastive decoding still make sense? Could the authors provide further clarifications on this point?
4. For some VQA scenarios, language priors play an essential role in generating appropriate answers. For example, when given an image of a celebrity and asked, “Please introduce this person in detail,” the LVLM only needs to identify who the person is from the image, and then relies heavily on language priors to provide a detailed introduction. Would the proposed method negatively affect the model’s ability to handle such cases where language priors are actually beneficial?

[a] Ibd: Alleviating hallucinations in large vision-language models via image-biased decoding, CVPR2025 Workshop

[b] Paying More Attention to Image: A Training-Free Method for Alleviating Hallucination in LVLMs, ECCV2024

**Questions:**

Please see the Weaknesses section.

---

### Official Review · Reviewer_dymS · 2025-10-28

**Soundness:** 3
**Presentation:** 2
**Contribution:** 2
**Rating:** 2
**Confidence:** 5

**Summary:**

This paper proposes a simple framework to reduce object hallucinations in LVLMs. The authors first analyze whether hallucinations originate from the vision encoder or the backbone LLM, finding that the language priors in the LLM are the main cause. To address this, they introduce NoLan, which contrasts the output distributions of multimodal and unimodal (text-only) inputs. NoLan comes in two variants: "NoLan-Base", which uses a fixed contrasting rate of 1, and "NoLan-Plus", which adaptively adjusts this rate based on KL divergence between the two distributions. Experiments across benchmarks show that NoLan significantly reduces hallucinations and outperforms prior training-free approaches like VCD, M3ID, and VDD. The framework is validated with various LVLMs.

**Strengths:**

* This paper addresses the important issue of hallucination in LVLMs.
* It demonstrates strong empirical performance.
* The proposed methodology is simple and easy to apply.

**Weaknesses:**

* The analysis that LVLM hallucinations are caused by language priors is not entirely new. Prior studies have already identified that hallucinations in LVLMs stem from a strong reliance on language priors [1, 2].
* The paper lacks methodological novelty, as it merely applies minor modifications to existing, extensively studied contrastive decoding-based approaches.
* Additionally, the proposed method introduces significant computational overhead at inference time, yet the paper does not include comparisons of computational cost with other approaches. Recent studies [3, 4] have shown that hallucinations can be substantially reduced with minimal additional computation, but this paper does not address such efficiency considerations.

[1] Leng et al., Mitigating Object Hallucinations in Large Vision-Language Models through Visual Contrastive Decoding, CVPR 2024.

[2] Min et al., Mitigating Hallucinations in Large Vision-Language Models via Summary-Guided Decoding, NAACL 2025 Findings.

[3] He et al., Cracking the Code of Hallucination in LVLMs with Vision-aware Head Divergence, ACL 2025.

[4] Xu et al., Mitigating Hallucinations in Multi-modal Large Language Models via Image Token Attention-Guided Decoding, NAACL 2025.

**Questions:**

* What if the original LLM is not available? How can we apply NoLan if the LVLM was trained from scratch without an LLM backbone, or if it was fine-tuned so extensively that its weights have significantly diverged from the original LLM?

---

### Official Review · Reviewer_NSgK · 2025-10-30

**Soundness:** 2
**Presentation:** 3
**Contribution:** 2
**Rating:** 4
**Confidence:** 3

**Summary:**

This paper targets the persistent issue of object hallucinations in Large Vision-Language Models (LVLMs)—instances where models generate text describing nonexistent objects. Through detailed experimentation, the authors identify that such hallucinations primarily originate from dominant language priors in the language decoder rather than deficiencies in the vision encoder. To mitigate this, they propose No-Language-Hallucination Decoding (NoLan), a simple, training-free method that dynamically adjusts the model’s output logits by comparing multimodal and text-only distributions. Extensive evaluations across diverse benchmarks demonstrate consistent reductions in hallucinations and enhanced grounding across multiple LVLM architectures.

**Strengths:**

1. The paper demonstrates that language priors are one of the primary sources of hallucination errors in LVLMs, rather than deficiencies in visual capability, and substantiates this viewpoint through detailed preliminary experiments.

2. The proposed NoLan framework is elegant, pragmatic, and computationally efficient compared to prior contrastive approaches such as VCD, M3ID, and VDD. The introduction of per-token, KL-divergence-based dynamic modulation in NoLan-Plus represents a more nuanced modeling of linguistic bias, enabling fine-grained suppression without retraining.

**Weaknesses:**

1. I believe that the notion of prior language knowledge is merely one of the illusions at play. As shown in Figure 3, the authors ask image-related questions directly after removing the image, which is not a fair approach. Even humans would not be able to answer correctly without access to the image.

2. The distribution of unimodal logits is extremely uncontrollable. As shown in Figure 1, when we ask about the most common animals, the presence of a large number of words not appearing in the image can severely impair the model’s performance.

3. This architecture is relatively slow, requiring two model calls to determine the output, which I believe will significantly affect inference speed.

**Questions:**

Please refer to Weaknesses

---

### Note · Authors · 2025-12-03

I have read and agree with the venue's withdrawal policy on behalf of myself and my co-authors.